# SQ Lower Bounds for Non-Gaussian Component Analysis with Weaker Assumptions

**Ilias Diakonikolas**
University of Wisconsin-Madison
ilias@cs.wisc.edu

**Daniel M. Kane**
University of California, San Diego
dakane@cs.ucsd.edu

**Lisheng Ren**
University of Wisconsin-Madison
lren29@wisc.edu

**Yuxin Sun**
University of Wisconsin-Madison
yxsun@cs.wisc.edu

## Abstract

We study the complexity of Non-Gaussian Component Analysis (NGCA) in the Statistical Query (SQ) model. Prior work developed a general methodology to prove SQ lower bounds for this task that have been applicable to a wide range of contexts. In particular, it was known that for any univariate distribution $A$ satisfying certain conditions, distinguishing between a standard multivariate Gaussian and a distribution that behaves like $A$ in a random hidden direction and like a standard Gaussian in the orthogonal complement, is SQ-hard. The required conditions were that (1) $A$ matches many low-order moments with the standard univariate Gaussian, and (2) the chi-squared norm of $A$ with respect to the standard Gaussian is finite. While the moment-matching condition is necessary for hardness, the chi-squared condition was only required for technical reasons. In this work, we establish that the latter condition is indeed not necessary. In particular, we prove near-optimal SQ lower bounds for NGCA under the moment-matching condition only. Our result naturally generalizes to the setting of a hidden subspace. Leveraging our general SQ lower bound, we obtain near-optimal SQ lower bounds for a range of concrete estimation tasks where existing techniques provide sub-optimal or even vacuous guarantees.

## 1 Introduction

Non-Gaussian Component Analysis (NGCA) is a statistical estimation task first considered in the signal processing literature [BKS+06]. As the name suggests, the objective is to find a non-gaussian direction (or, more generally, low-dimensional subspace) in a high-dimensional dataset. Since its introduction, the NGCA problem has been studied in a range of works from an algorithmic standpoint; see [TV18, GS19] and references therein. Here we explore this problem from a hardness perspective in the Statistical Query (SQ) model. Before we motivate and state our results, we require basic background on the SQ model.

**SQ Model**  SQ algorithms are a class of algorithms that are allowed to query expectations of bounded functions on the underlying distribution through the SQ oracle rather than directly access samples. The model was introduced by Kearns [Kea98] as a natural restriction of the PAC model [Val84] in the context of learning Boolean functions. Since then, the SQ model has been extensively studied in a range of settings, including unsupervised learning [Fel16]. The class of SQ algorithms is broad and captures a range of known algorithmic techniques in machine learning in-

37th Conference on Neural Information Processing Systems (NeurIPS 2023).

cluding spectral techniques, moment and tensor methods, local search (e.g., EM), and many others (see, e.g., [FGR$^+$17, FGV17] and references therein).

**Definition 1.1** (SQ Model). Let $D$ be a distribution on $\mathbb{R}^n$. A *statistical query* is a bounded function $q : \mathbb{R}^n \to [-1, 1]$. We define $\mathrm{STAT}(\tau)$ to be the oracle that given any such query $q$, outputs a value $v$ such that $|v - \mathbf{E}_{\mathbf{x} \sim D}[q(\mathbf{x})]| \leq \tau$, where $\tau > 0$ is the *tolerance* parameter of the query. A *statistical query (SQ) algorithm* is an algorithm whose objective is to learn some information about an unknown distribution $D$ by making adaptive calls to the corresponding $\mathrm{STAT}(\tau)$ oracle.

The following family of high-dimensional distributions forms the basis for the definition of the NGCA problem.

**Definition 1.2** (Hidden-Subspace Distribution). For a distribution $A$ supported on $\mathbb{R}^m$ and a matrix $\mathbf{V} \in \mathbb{R}^{n \times m}$ with $\mathbf{V}^\intercal \mathbf{V} = \mathbf{I}_m$, we define the distribution $\mathbf{P}_{\mathbf{V}}^A$ supported on $\mathbb{R}^n$ such that it is distributed according to $A$ in the subspace $\mathrm{span}(\mathbf{v}_1, \ldots, \mathbf{v}_m)$ and is an independent standard Gaussian in the orthogonal directions, where $\mathbf{v}_1, \ldots, \mathbf{v}_m$ denote the column vectors of $\mathbf{V}$. In particular, if $A$ is a continuous distribution with probability density function $A(\mathbf{y})$, then $\mathbf{P}_{\mathbf{V}}^A$ is the distribution over $\mathbb{R}^n$ with probability density function

$$\mathbf{P}_{\mathbf{V}}^A(\mathbf{x}) = A(\langle \mathbf{v}_1, \mathbf{x} \rangle, \ldots, \langle \mathbf{v}_m, \mathbf{x} \rangle) \exp(-\|\mathbf{x} - \mathbf{V}\mathbf{V}^\intercal \mathbf{x}\|_2^2/2)/(2\pi)^{(n-m)/2} \ .$$

That is, $\mathbf{P}_{\mathbf{V}}^A$ is the product distribution whose orthogonal projection onto the subspace of $\mathbf{V}$ is $A$, and onto the subspace perpendicular to $\mathbf{V}$ is the standard $(n - m)$-dimensional normal distribution. An important special case of the above definition considers to $m = 1$ (i.e., the case when the hidden subspace is a hidden direction); for this setting, we will use the notation $\mathbf{P}_{\mathbf{v}}^A$ for such a distribution, where $A$ is a one-dimensional distribution and $\mathbf{v} \in \mathbb{R}^n$ is a unit vector.

Since we are focusing on establishing hardness, we will consider the following hypothesis testing version of NGCA (since the learning/search version typically reduces to the testing problem). We use $\mathcal{N}_n$ to denote the standard $n$-dimensional Gaussian distribution $\mathcal{N}(0, \mathbf{I}_n)$. We use $U(\mathbf{O}_{n,m})$ to denote the uniform distribution over the set of all orthogonal matrices $\mathbf{V} \in \mathbb{R}^{n \times m}$; namely, this is the distribution obtained by taking $\mathbf{R}\mathbf{V}'$, where $\mathbf{R} \in \mathbb{R}^{n \times n}$ is a random rotation matrix and $\mathbf{V}' \in \mathbb{R}^{n \times m}$ is an arbitrary orthogonal matrix.

**Definition 1.3** (Hypothesis Testing Version of NGCA). Let $n > m \geq 1$ be integers. For a distribution $A$ supported on $\mathbb{R}^m$, one is given access to a distribution $D$ such that either:

- $H_0$: $D = \mathcal{N}_n$,

- $H_1$: $D$ is given by $\mathbf{P}_{\mathbf{V}}^A$, where $\mathbf{V} \sim U(\mathbf{O}_{n,m})$.

The goal is to distinguish between these two cases $H_0$ and $H_1$.

For the special case that $m = 1$ (i.e., for a univariate distribution $A$), prior work [DKS17] established SQ-hardness of NGCA[1] under the following condition:

**Condition 1.4.** *Let $d \in \mathbb{Z}_+$. The distribution $A$ on $\mathbb{R}$ is such that (i) the first $d$ moments of $A$ agree with the first $d$ moments of $\mathcal{N}(0, 1)$, and (ii) the chi-squared distance $\chi^2(A, \mathcal{N})$ is finite, where the chi-squared distance for two distributions (with probability density functions) $P, Q : \mathbb{R}^n \to \mathbb{R}_+$ is defined as $\chi^2(P, Q) \stackrel{\text{def}}{=} \int_{\mathbf{x} \in \mathbb{R}^n} P(\mathbf{x})^2/Q(\mathbf{x})d\mathbf{x} - 1$.*

Specifically, the main result of [DKS17] shows that any SQ algorithm that solves the testing version of NGCA requires either $2^{n^{\Omega(1)}}$ many SQ queries or at least one query with accuracy

$$n^{-\Omega(d)}\sqrt{\chi^2(A, \mathcal{N}(0, 1))}.$$

It is worth noting that subsequent works (see [DK22a] and [DKPZ21]) generalized this result so that it only requires that (i) $A$ *approximately* matches moments with the standard Gaussian, and (ii) $A$ is a low-dimensional distribution embedded in a hidden low-dimensional subspace, instead of a one-dimensional distribution.

---

[1]While the SQ lower bound result of [DKS17] was phrased for the search version of NGCA, they can be directly translated to the testing version; see, e.g., Chapter 8 of [DK23].

The starting point of our investigation is a key technical limitation of this line of work. Specifically, if $\chi^2(A, \mathcal{N}(0,1))$ is very large (or infinite), e.g., if $A$ has constant probability mass on a discrete set, the aforementioned SQ lower bound of [DKS17] can be very weak (or even vacuous). It is thus natural to ask if the finite chi-squared assumption is in fact necessary for the corresponding SQ lower bounds to hold.

A concrete motivation to answer this question comes from the applications of a generic SQ lower bound for NGCA to various learning problems. The SQ-hardness of NGCA can be used to obtain similar hardness for a number of well-studied learning problems that superficially appear very different. These include learning mixture models [DKS17, DKPZ23, DKS23], robust mean/covariance estimation [DKS17], robust linear regression [DKS19], learning halfspaces and other natural concepts with adversarial or semi-random label noise [DKZ20, GGK20, DK22a, DKPZ21, DKK+22], list-decodable mean estimation and linear regression [DKS18, DKP+21], learning simple neural networks [DKKZ20, GGJ+20], and even learning simple generative models [CLL22]. In several of these applications, the requirement of bounded chi-squared distance is somewhat problematic, in some cases leading to quantitatively sub-optimal results. Moreover, in certain applications, this restriction leads to vacuous guarantees.

## 1.1 Our Results

### 1.1.1 Main Result

Our main result is a qualitatively near-optimal (up to the constant factor in the exponent) SQ lower bound for the NGCA problem, assuming only the moment-matching condition (i.e., without the chi-squared distance restriction). Informally, we essentially show that in order to solve the NGCA in $n$ dimensions with an $m$-dimensional distribution $A$ that (approximately) matches moments with the standard Gaussian up to degree $d$, any SQ algorithm will either require one query with accuracy $O_{m,d}(n^{-\Omega(d)})$ or exponential in $n$ many queries.

Formally, we establish the following theorem.

**Theorem 1.5** (Main SQ Lower Bound Result). *Let $\lambda \in (0,1)$ and $n, m, d \in \mathbb{N}$ with $d$ even and $m, d \leq n^\lambda$. Let $\nu \in \mathbb{R}_+$ and $A$ be a distribution on $\mathbb{R}^m$ such that for any polynomial $f : \mathbb{R}^m \to \mathbb{R}$ of degree at most $d$ and $\mathbf{E}_{\mathbf{x} \sim \mathcal{N}_m}[f(\mathbf{x})^2] = 1$, the following holds: $|\mathbf{E}_{\mathbf{x} \sim A}[f(\mathbf{x})] - \mathbf{E}_{\mathbf{x} \sim \mathcal{N}_m}[f(\mathbf{x})]| \leq \nu$. Let $0 < c < (1-\lambda)/8$ and $n$ be at least a sufficiently large constant depending on $c$. Then any SQ algorithm solving the $n$-dimensional NGCA problem (as in Definition 1.3) with $2/3$ success probability requires either (i) a query of tolerance $O_{m,d}\left(n^{-((1-\lambda)/8-c)d}\right) + (1 + o(1))\nu$, or (ii) $2^{n^{\Omega(c)}}$ many queries.*

A few comments regarding the parameters are in order here. We first note that the constant in $O_{m,d}(n^{-((1-\lambda)/8-c)d})$ is roughly the size of the Binomial coefficient $\binom{(d+m)/2-1}{m/2-1}$. Furthermore, we would like to point out that for most applications we will have $m, d < 1/\mathrm{poly}(n)$. Therefore, the parameters $\lambda$ and $c$ can be taken to be arbitrarily close to 0. So, informally, our lower bound on the query accuracy can be roughly thought of as $\binom{(d+m)/2-1}{m/2-1}n^{-d/8}$.

For all the applications given in this paper, the above theorem will be applied for the special case that $m = 1$; namely, the case that the hidden subspace is a hidden direction and $A$ is a univariate distribution.

**Relation to LLL-based Algorithms for NGCA** Consider the special case of the NGCA problem corresponding to $m = 1$ where $A$ is a *discrete* distribution that matches its first $d$ moments with the standard Gaussian. Theorem 1.5 implies that any SQ algorithm for this version of the problem either uses a query with accuracy $n^{-\Omega(d)}$ or exponential many queries. On the other hand, recent works[DK22b, ZSWB22] gave polynomial-time algorithms for this problem with sample complexity $O(n)$, regardless of the degree $d$. It is worth noting that the existence of these algorithms does not contradict our SQ lower bound, as these algorithms are based on the LLL-method for lattice basis reduction that is not captured by the SQ framework. An an implication, it follows that LLL-based methods surpass any efficient SQ algorithm for these settings of NGCA. A similar observation was previously made in [DH23] for the special case that the discrete distribution $A$ is supported on $\{0, \pm 1\}$; consequently, [DH23] could only obtain a quadratic separation. Finally, we note that this

limitation of SQ algorithms is also shared by two other prominent restricted families of algorithms (namely, SoS algorithms and low-degree polynomial tests).

### 1.1.2 Applications

We believe that Theorem 1.5 is interesting in its own right, as it elucidates the SQ complexity of a natural and well-studied statistical problem. Here we discuss concrete applications to some natural statistical tasks. We note that the SQ lower bound in the prior work [DKS17] cannot give optimal (or even nontrivial) lower bounds for these applications.

**List-decodable Gaussian Mean Estimation**    One can leverage our result to obtain a sharper SQ lower bound for the task of list-decodable Gaussian mean estimation. In this task, the algorithm is given as input points from $\mathbb{R}^n$ where an $\alpha < 1/2$ fraction of the points are drawn from an unknown mean and identity covariance Gaussian $\mathcal{N}(\boldsymbol{\mu}, \mathbf{I})$, and the remaining points are arbitrary. The goal of the algorithm is to output a list of $O(1/\alpha)$ many hypothesis vectors at least one of which is close to $\boldsymbol{\mu}$ in $\ell_2$-norm with probability at least $2/3$. [DKS18] established the following SQ lower bound for this problem (see also [DK23] for a different exposition).

**Fact 1.6** ([DKS18])**.** *For each $d \in \mathbb{Z}_+$ and $c \in (0, 1/2)$, there exists $c_d > 0$ such that for any $\alpha > 0$ sufficiently small the following holds. Any SQ algorithm that is given access to a $(1 - \alpha)$-corrupted Gaussian $\mathcal{N}(\boldsymbol{\mu}, \mathbf{I})$ in $n > d^{3/c}$ dimensions and returns a list of hypotheses such that with probability at least $2/3$ one of the hypotheses is within $\ell_2$-distance $c_d \alpha^{-1/d}$ of the true mean $\boldsymbol{\mu}$, does one of the following: (i) Uses queries with error tolerance at most $\exp(O(\alpha^{-2/d}))\Omega(n)^{-(d+1)(1/4-c/2)}$. (ii) Uses at least $\exp(\Omega(n^c))$ many queries. (iii) Returns a list of at least $\exp(\Omega(n))$ many hypotheses.*

The above statement is obtained using the framework of [DKS17] by considering the distribution testing problem between $\mathcal{N}(\boldsymbol{\mu}, \mathbf{I})$ and $\mathbf{P}_{\mathbf{v}}^A$, where $A$ is the one-dimensional moment-matching distribution of the following form.

**Fact 1.7** ([DKS18], see Lemma 8.21 in [DK23])**.** *For $d \in \mathbb{Z}_+$, there exists a distribution $A = \alpha \mathcal{N}(\mu, 1) + (1 - \alpha)E$, for some distribution $E$ and $\mu = 10 c_d \alpha^{-1/d}$, such that the first $d$ moments of $A$ agree with those of $\mathcal{N}(0, 1)$. Furthermore, the probability density function of $E$ can be taken to be pointwise at most twice the pdf of the standard Gaussian.*

We note that $\chi^2(A, \mathcal{N}) = O(\exp(\mu^2))$, and this results in the $\exp(O(\alpha^{-2/d}))$ term in the error tolerance of Fact 1.6. Consequently, if $\alpha \ll \log(n)^{-d/2}$, the error tolerance is greater than one and Fact 1.6 fails to give a non-trivial bound.   It is worth noting that the setting of "small $\alpha$" (e.g., $\alpha$ is sub-constant in the dimension) is of significant interest in various applications, including in mean estimation. A concrete application is in the related crowd-sourcing setting of [MV18] dealing with this parameter regime.

We can circumvent this technical problem by combining our main result (Theorem 1.5) with Fact 1.7 to obtain the following sharper SQ lower bound (see Appendix C for the proof).

**Theorem 1.8** (SQ Lower Bound for List-Decoding the Mean)**.** *Let $\lambda \in (0, 1)$ and $n, d \in \mathbb{N}$ with $d$ be even and $d \leq n^\lambda$. Let $c > 0$ and $n$ be at least a sufficiently large constant depending on $c$. There exists $c_d > 0$ such that for any $\alpha > 0$ sufficiently small the following holds. Any SQ algorithm that is given access to a $(1 - \alpha)$-corrupted Gaussian $\mathcal{N}(\boldsymbol{\mu}, \mathbf{I})$ in $n$ dimensions and returns a list of hypotheses such that with probability at least $2/3$ one of the hypotheses is within $\ell_2$-distance $c_d \alpha^{-1/d}$ of the true mean $\boldsymbol{\mu}$, does one of the following: (i) Uses queries with error tolerance at most $O_d(n^{-((1-\lambda)/8-c)d})$. (ii) Uses at least $2^{n^{\Omega(c)}}$ many queries. (iii) Returns a list of at least $\exp(\Omega(n))$ many hypotheses.*

**Anti-concentration Detection**    Anti-concentration (AC) detection is the following hypothesis testing problem: given access to an unknown distribution $D$ over $\mathbb{R}^n$, where the input distribution $D$ is promised to satisfy either (i) $D$ is the standard Gaussian; or (ii) $D$ has at least $\alpha < 1$ of its probability mass residing inside a dimension $(n - 1)$ subspace $V \subset \mathbb{R}^n$. The goal is to distinguish between the two cases with success probability $2/3$.

In order to use our main result to derive an SQ lower bound for this task, we require the following lemma on univariate moment-matching (see Appendix C for the proof).

**Lemma 1.9.** *Let $D_0$ denote the distribution that outputs $0$ with probability $1$. For $d \in \mathbb{Z}_+$, there exists $\alpha_d \in (0,1)$ and a univariate distribution $A = \alpha_d D_0 + (1 - \alpha_d) E$ for some distribution $E$, such that the first $d$ moments of $A$ agree with those of $\mathcal{N}(0,1)$. Furthermore, the pdf of $E$ can be taken to be pointwise at most twice the pdf of the standard Gaussian.*

Using the above lemma and our main result, we can deduce the following SQ lower bound on the anti-concentration detection problem (see Appendix C for the proof).

**Theorem 1.10** (SQ Lower Bound for AC Detection). *For any $\alpha \in (0, 1/2)$, any SQ algorithm that has access to a $n$-dimensional distribution that is either (i) the standard Gaussian; or (ii) a distribution that has at least $\alpha < 1$ probability mass in a (n-1)-dimensional subspace $V \subset \mathbb{R}^n$, and distinguishes the two cases with success probability at least $2/3$, either requires a query with error at most $n^{-\omega_\alpha(1)}$, or uses at least $2^{n^{\Omega(1)}}$ many queries.*

**Learning Periodic Functions** Another application is for the well-studied problem of learning *periodic* functions, see, e.g., [SVWX17] and [SZB21], which is closely related to the continuous Learning with Errors (cLWE) problem [BRST21]. In the task of learning periodic functions, the algorithm is given sample access to a distribution $D$ of $(\mathbf{x}, y)$ over $\mathbb{R}^n \times \mathbb{R}$. The distribution $D$ is such that $\mathbf{x} \sim \mathcal{N}(0, \mathbf{I}_n)$ and $y = \cos(2\pi(\delta\langle\mathbf{w}, \mathbf{x}\rangle + \zeta))$ with noise $\zeta \sim \mathcal{N}(0, \sigma^2)$. This implies that $y$ is a periodic function of $\mathbf{x}$ along the direction of $\mathbf{w}$ with a small amount of noise. While the frequency and noise scale parameters $\delta, \sigma \in \mathbb{R}_+$ are known to the algorithm, the parameter $\mathbf{w} \in \mathbb{S}^{n-1}$ is unknown; the goal of the algorithm is to output a hypothesis $h : \mathbb{R}^n \to \mathbb{R}$ such that $\mathbf{E}_{\mathbf{x} \sim D}[(h(\mathbf{x}) - y)^2]$ is minimized.

In order to use our main theorem to derive an SQ lower bound for this problem, we need to show that for any $t \in [-1, 1]$, $D$ conditioned on $y = t$ approximately matches moments with $\mathcal{N}(0, \mathbf{I}_n)$. To this end, we introduce the following definition and fact of discrete Gaussian measure from [DK22a].

**Definition 1.11.** *For $s \in \mathbb{R}_+$ and $\theta \in \mathbb{R}$, let $G_{s,\theta}$ denote the measure of the "$s$-spaced discrete Gaussian distribution". In particular, for each $n \in \mathbb{Z}$, $G_{s,\theta}$ assigns mass $sg(ns + \theta)$ to the point $ns + \theta$, where $g$ is the pdf function of $\mathcal{N}(0,1)$.*

Note that although $G_{s,\theta}$ is not a probability measure (as the total measure is not one), it can be thought of as a probability distribution since the total measure is close to one for small $\sigma$. To see this, we introduce the following fact.

**Fact 1.12** (Lemma 3.12 from [DK22a]). *For all $k \in \mathbb{N}$, $s > 0$ and all $\theta \in \mathbb{R}$, we have that $|\mathbf{E}_{t \sim \mathcal{N}(0,1)}[t^k] - \mathbf{E}_{t \sim G_{s,\theta}}[t^k]| = k!O(s)^k \exp(-\Omega(1/s^2))$.*

Using the above fact, we are now ready to prove our SQ lower bound for learning periodic functions (see Appendix C for the proof).

**Theorem 1.13** (SQ Lower Bound for Learning Periodic Functions). *Let $c > 0$ and $n$ be at least a sufficiently large constant depending on $c$. Let $D$ be the distribution of $(\mathbf{x}, y)$ over $\mathbb{R}^n \times \mathbb{R}$ that is generated by $\mathbf{x} \sim \mathcal{N}(0, \mathbf{I}_n)$ and $y = \cos(2\pi(\delta\langle\mathbf{w}, \mathbf{x}\rangle + \zeta))$ with noise $\zeta \sim \mathcal{N}(0, \sigma^2)$. Let $\delta > n^c$, $\sigma$ be known and $\mathbf{w}$ be unknown to the algorithm. Then any SQ algorithm that has access to the distribution $D$ and returns a hypothesis $h : \mathbb{R}^n \to \mathbb{R}$ such that $\mathbf{E}_{(\mathbf{x},y) \sim D}[(h(\mathbf{x}) - y)^2] = o(1)$, either requires a query with error at most $\exp(-n^{c'})$ for $c' < \min(2c, 1/10)$, or uses at least $2^{n^{\Omega(1)}}$ many queries.*

## 1.2 Technical Overview

We start by noting that we cannot use the standard SQ dimension argument [FGR+17] to prove our result, due to the unbounded chi-squared norm. To handle this issue, we need to revisit the underlying ideas of that proof. In particular, we will show that for any bounded query function $f : \mathbb{R}^n \to [-1, 1]$, with high probability over some $\mathbf{V} \sim U(\mathbf{O}_{n,m})$, it holds that $|\mathbf{E}_{\mathbf{x} \sim \mathcal{N}_n}[f(\mathbf{x})] - \mathbf{E}_{\mathbf{x} \sim \mathbf{P}_\mathbf{V}^A}[f(\mathbf{x})]|$ will be small. This allows an adversarial oracle to return $\mathbf{E}_{\mathbf{x} \sim \mathcal{N}_n}[f(\mathbf{x})]$ to every query $f$ regardless of which case we are in, unless the algorithm is lucky enough (or uses high accuracy) to find an $f$ that causes $|\mathbf{E}_{\mathbf{x} \sim \mathcal{N}_n}[f(\mathbf{x})] - \mathbf{E}_{\mathbf{x} \sim \mathbf{P}_\mathbf{V}^A}[f(\mathbf{x})]|$ to be at least $\tau$.

Our approach for calculating $\mathbf{E}_{\mathbf{x} \sim \mathbf{P}_\mathbf{V}^A}[f(\mathbf{x})]$ will be via Fourier analysis. In particular, using the Fourier decomposition of $f$, we can write $f(\mathbf{x}) = \sum_{k=0}^{\infty} \langle \mathbf{T}_k, \mathbf{H}_k(\mathbf{x}) \rangle$, where $\mathbf{H}_k(\mathbf{x})$ is the

properly normalized degree-$k$ Hermite polynomial tensor and $\mathbf{T}_k$ is the degree-$k$ Fourier coefficients of $f$. Taking the inner product with the distribution $\mathbf{P}_{\mathbf{V}}^A$ involves computing the expectation of $\mathbf{E}_{\mathbf{x} \sim \mathbf{P}_{\mathbf{V}}^A}[\mathbf{H}_k(\mathbf{x})]$, which can be seen to be $\langle \mathbf{V}^{\otimes k} \mathbf{A}_k, \mathbf{T}_k \rangle$, where $\mathbf{A}_k = \mathbf{E}_{\mathbf{x} \sim A}[\mathbf{H}_k(\mathbf{x})]$. Thus, we obtain (at least morally speaking) that

$$\mathbf{E}_{\mathbf{x} \sim \mathbf{P}_{\mathbf{V}}^A}[f(\mathbf{x})] = \sum_{k=0}^{\infty} \langle \mathbf{V}^{\otimes k} \mathbf{A}_k, \mathbf{T}_k \rangle \le \sum_{k=0}^{\infty} |\langle \mathbf{A}_k, (\mathbf{V}^{\mathsf{T}})^{\otimes k} \mathbf{T}_k \rangle| \le \sum_{k=0}^{\infty} \|\mathbf{A}_k\|_2 \|(\mathbf{V}^{\mathsf{T}})^{\otimes k} \mathbf{T}_k\|_2. \quad (1)$$

The $k = 0$ term of the above sum will be exactly $\mathbf{T}_0 = \mathbf{E}_{\mathbf{x} \sim \mathcal{N}_n}[f(\mathbf{x})]$. By the moment-matching condition, the $k = 1$ through $k = d$ terms will be very small. Finally, we need to argue that the higher degree terms are small with high probability. To achieve this, we provide a non-trivial upper bound for $\mathbf{E}_{\mathbf{V} \sim U(\mathbf{O}_{n,m})}[\|(\mathbf{V}^{\mathsf{T}})^{\otimes k} \mathbf{T}_k\|_2^2]$. Combining with the fact that $\sum_{k=0}^{\infty} \|\mathbf{T}_k\|_2^2 = \|f\|_2^2 \le 1$, this allows us to prove high probability bounds on each term, and thus their sum.

Furthermore, if we want higher probability estimates of the terms with small $k$, we can instead bound $\mathbf{E}_{\mathbf{V} \sim U(\mathbf{O}_{n,m})}[\|(\mathbf{V}^{\mathsf{T}})^{\otimes k} \mathbf{T}_k\|_2^{2a}]$ for some integer $a$. Unfortunately, there are non-trivial technical issues with the above approach, arising from issues with (1). To begin with, as no assumptions other than moment-matching (for a few low-degree moments) were made on $A$, it is not guaranteed that $\mathbf{A}_k$ is finite for larger values of $k$. To address this issue, we will truncate the distribution $A$. In particular, we pick a parameter $B$ (which will be determined carefully), and define $A'$ to be $A$ conditioned on the value in the ball $\mathbb{B}^m(B)$. Due to the higher-moment bounds, we can show that $A$ and $A'$ are close in total variation distance, and thus that $\mathbf{E}_{\mathbf{x} \sim \mathbf{P}_{\mathbf{V}}^A}[f(\mathbf{x})]$ is close to $\mathbf{E}_{\mathbf{x} \sim \mathbf{P}_{\mathbf{V}}^{A'}}[f(\mathbf{x})]$ for any bounded $f$.

Furthermore, using the higher moment bounds, we can show that $A'$ *nearly* matches the low-degree moments of $\mathcal{N}_m$. The second issue arises with the interchange of summations used to derive (1). In particular, although $f(\mathbf{x}) = \sum_{k=0}^{\infty} \langle \mathbf{T}_k, \mathbf{H}_k(\mathbf{x}) \rangle$, it does not necessarily follow that we can interchange the infinite sum on the right-hand-side with taking the expectation over $\mathbf{x} \sim \mathbf{P}_{\mathbf{V}}^A$. To fix this issue, we split $f$ into two parts $f^{\le \ell}$ (consisting of its low-degree Fourier components) and $f^{> \ell}$. We note that (1) *does* hold for $f^{\le \ell}$, as the summation there will be finite, and we can use the above argument to bound $\mathbf{E}_{\mathbf{x} \sim \mathcal{N}_n}[f(\mathbf{x})] - \mathbf{E}_{\mathbf{x} \sim \mathbf{P}_{\mathbf{V}}^A}[f^{\le \ell}(\mathbf{x})]|$ with high probability. To bound $|\mathbf{E}_{\mathbf{x} \sim \mathbf{P}_{\mathbf{V}}^A}[f^{> \ell}(\mathbf{x})]|$, we note that by taking $\ell$ large, we can make $\|f^{> \ell}\|_2 < \delta$ for some exponentially small $\delta > 0$. We then bound $\mathbf{E}_{\mathbf{V} \sim U(\mathbf{O}_{n,m})}[\mathbf{E}_{\mathbf{x} \sim \mathbf{P}_{\mathbf{V}}^A}[|f^{> \ell}|]] = \mathbf{E}_{\mathbf{x} \sim \mathbf{Q}}[|f^{> \ell}|]$, where $\mathbf{Q}$ is the average over $\mathbf{V}$ of $\mathbf{P}_{\mathbf{V}}^A$ (note that everything here is non-negative, so there is no issue with the interchange of integrals). Thus, we can bound the desired quantity by noting that $\|f^{> \ell}\|_2$ is small and that the chi-squared norm of $\mathbf{Q}$ with respect to the standard Gaussian $\mathcal{N}_n$ is bounded.

## 2  Preliminaries

We will use lowercase boldface letters for vectors and capitalized boldface letters for matrices and tensors. We use $\mathbb{S}^{n-1} = \{\mathbf{x} \in \mathbb{R}^n : \|\mathbf{x}\|_2 = 1\}$ to denote the $n$-dimensional unit sphere. For vectors $\mathbf{u}, \mathbf{v} \in \mathbb{R}^n$, we use $\langle \mathbf{u}, \mathbf{v} \rangle$ to denote the standard inner product. For $\mathbf{u} \in \mathbb{R}^n$, we use $\|\mathbf{u}\|_k = \left( \sum_{i=1}^{n} \mathbf{u}_i^k \right)^{1/k}$ to denote the $\ell^k$-norm of $\mathbf{u}$. For tensors, we will consider a $k$-tensor to be an element in $(\mathbb{R}^n)^{\otimes k} \cong \mathbb{R}^{n^k}$. This can be thought of as a vector with $n^k$ coordinates. We will use $\mathbf{A}_{i_1, \dots, i_k}$ to denote the coordinate of a $k$-tensor $\mathbf{A}$ indexed by the $k$-tuple $(i_1, \dots, i_k)$. By abuse of notation, we will sometimes also use this to denote the entire tensor. The inner product and $\ell^k$-norm of $k$-tensor are defined by thinking of the tensor as vectors with $n^k$ coordinates and then use the definition of inner product and $\ell^k$-norm of vectors. For a vector $\mathbf{v} \in \mathbb{R}^n$, we denote by $\mathbf{v}^{\otimes k}$ to be a vector (linear object) in $\mathbb{R}^{n^k}$. For a matrix $\mathbf{V} \in \mathbb{R}^{n \times m}$, we denote by $\|\mathbf{V}\|_2, \|\mathbf{V}\|_F$ to be the operator norm and Frobenius norm respectively. In addition, we denote by $\mathbf{V}^{\otimes k}$ to be a matrix (linear operator) mapping $\mathbb{R}^{n^k}$ to $\mathbb{R}^{m^k}$. We use $\mathbb{1}$ to denote the indicator function of a set, specifically $\mathbb{1}(t \in S) = 1$ if $t \in S$ and 0 otherwise. We will use $\Gamma : \mathbb{R} \to \mathbb{R}$ to denote the gamma function $\Gamma(z) = \int_0^{\infty} t^{z-1} e^{-t} dt$. We use $B : \mathbb{R} \times \mathbb{R} \to \mathbb{R}$ to denote the beta function $B(z_1, z_2) = \Gamma(z_1) \Gamma(z_2) / \Gamma(z_1 + z_2)$. We use $\chi_k^2$ to denote the chi-squared distribution with $k$ degrees of freedom. We use $\text{Beta}(\alpha, \beta)$ to denote the Beta distribution with parameters $\alpha$ and $\beta$.

For a distribution $D$, we use $\mathbf{Pr}_D[S]$ to denote the probability of an event $S$. For a continuous distribution $D$ over $\mathbb{R}^n$, we sometimes use $D$ for both the distribution itself and its probability density function. For two distributions $D_1, D_2$ over a probability space $\Omega$, let $d_{\mathrm{TV}}(D_1, D_2) = \sup_{S \subseteq \Omega} |\mathbf{Pr}_{D_1}(S) - \mathbf{Pr}_{D_2}(S)|$ denote the total variation distance between $D_1$ and $D_2$. For two continuous distribution $D_1, D_2$ over $\mathbb{R}^n$, we use $\chi^2(D_1, D_2) = \int_{\mathbb{R}^n} D_1(\mathbf{x})^2/D_2(\mathbf{x})d\mathbf{x} - 1$ to denote the chi-square norm of $D_1$ w.r.t. $D_2$. For a subset $S \subseteq \mathbb{R}^n$ with finite measure or finite surface measure, we use $U(S)$ to denote the uniform distribution over $S$ (w.r.t. Lebesgue measure for the volumn/surface area of $S$).

**Basics of Hermite Polynomials**

**Definition 2.1** (Normalized Hermite Polynomial). For $k \in \mathbb{N}$, we define the $k$-th *probabilist's* Hermite polynomials $He_k : \mathbb{R} \to \mathbb{R}$ as $He_k(t) = (-1)^k e^{t^2/2} \cdot \frac{d^k}{dt^k} e^{-t^2/2}$. We define the $k$-th *normalized* Hermite polynomial $h_k : \mathbb{R} \to \mathbb{R}$ as $h_k(t) = He_k(t)/\sqrt{k!}$.

Furthermore, we will use multivariate Hermite polynomials in the form of Hermite tensors (as the entries in the Hermite tensors are rescaled multivariate Hermite polynomials). We define the *Hermite tensor* as follows.

**Definition 2.2** (Hermite Tensor). For $k \in \mathbb{N}$ and $\mathbf{x} \in \mathbb{R}^n$, we define the $k$-th Hermite tensor as

$$(\mathbf{H}_k(\mathbf{x}))_{i_1,i_2,\ldots,i_k} = \frac{1}{\sqrt{k!}} \sum_{\substack{\text{Partitions } P \text{ of } [k] \\ \text{into sets of size 1 and 2}}} \bigotimes_{\{a,b\}\in P} (-\mathbf{I}_{i_a,i_b}) \bigotimes_{\{c\}\in P} \mathbf{x}_{i_c} .$$

We denote by $L^2(\mathbb{R}^n, \mathcal{N}_n)$ the function space of all functions $f : \mathbb{R}^n \to \mathbb{R}$ such that $\mathbf{E}_{\mathbf{v}\sim\mathcal{N}_n}[f^2(\mathbf{v})] < \infty$. For functions $f, g \in L^2(\mathbb{R}^n, \mathcal{N}_n)$, we use $\langle f, g \rangle_{\mathcal{N}_n} = \mathbf{E}_{\mathbf{x}\sim\mathcal{N}_n}[f(\mathbf{x})g(\mathbf{x})]$ to denote their inner product. We use $\|f\|_2 = \sqrt{\langle f, f \rangle_{\mathcal{N}_n}}$ to denote its $L^2$-norm. For a function $f : \mathbb{R}^n \to \mathbb{R}$ and $\ell \in \mathbb{N}$, we use $f^{\leq \ell}$ to denote $f^{\leq \ell}(\mathbf{x}) = \sum_{k=0}^\ell \langle \mathbf{A}_k, \mathbf{H}_k(\mathbf{x}) \rangle$, where $\mathbf{A}_k = \mathbf{E}_{\mathbf{x}\sim\mathcal{N}_n}[f(\mathbf{x})\mathbf{H}_k(\mathbf{x})]$, which is the degree-$\ell$ approximation of $f$. We use $f^{>\ell} = f - f^{\leq \ell}$ to denote its residue. We remark that normalized Hermite polynomials (resp. Hermite tensors) form a complete orthogonal system for the inner product space $L^2(\mathbb{R}, \mathcal{N})$ (resp. $L^2(\mathbb{R}^n, \mathcal{N}_n)$). This implies that for $f \in L^2(\mathbb{R}^n, \mathcal{N}_n)$, $\lim_{\ell\to\infty} \|f^{>\ell}\|_2 = 0$. We also remark that both our definition of Hermite polynomial and Hermite tensor are "normalized". In the sense that for Hermite polynomials, $\|h_k\|_2 = 1$. For Hermite tensors, given any symmetric tensor $A$, we have $\|\langle \mathbf{A}, \mathbf{H}_k(\mathbf{x}) \rangle\|_2^2 = \langle \mathbf{A}, \mathbf{A} \rangle$. The following claim states that for any orthonormal transformation $\mathbf{B}$, the Hermite tensor $\mathbf{H}_k(\mathbf{Bx})$ can be written as applying the linear transformation $\mathbf{B}^{\otimes k}$ on the Hermite tensor $\mathbf{H}_k(\mathbf{x})$. The proof is deferred to Appendix A.

**Claim 2.3.** *Let $1 \leq m < n$. Let $\mathbf{B} \in \mathbb{R}^{m \times n}$ with $\mathbf{BB}^\mathsf{T} = \mathbf{I}_m$. It holds that $\mathbf{H}_k(\mathbf{Bx}) = \mathbf{B}^{\otimes k} \mathbf{H}_k(\mathbf{x}), \mathbf{x} \in \mathbb{R}^n$.*

## 3 SQ-Hardness of NGCA: Proof of Theorem 1.5

The main idea of the proof is the following. Suppose that the algorithm only asks queries with tolerance $\tau$, and let $f$ be an arbitrary query function that the algorithm selects. The key ingredient is to show that $|\mathbf{E}_{\mathbf{x}\sim\mathbf{P}_{\mathbf{V}}^A}[f(\mathbf{x})] - \mathbf{E}_{\mathbf{x}\sim\mathcal{N}_n}[f(\mathbf{x})]| \leq \tau$ with high probability over $\mathbf{V} \sim U(\mathbf{O}_{n,m})$. If this holds, then when the algorithm queries $f$, if the input is from the alternative hypothesis, with high probability, $\mathbf{E}_{\mathbf{x}\sim\mathcal{N}_n}[f(\mathbf{x})]$ is a valid answer for the query. Therefore, when the algorithm queries $f$, regardless of whether the input is from the alternative or null hypothesis, the oracle can just return $\mathbf{E}_{\mathbf{x}\sim\mathcal{N}_n}[f(\mathbf{x})]$. Then the algorithm will not observe any difference between the two cases with any small number of queries. Thus, it is impossible to distinguish the two cases with high probability. To prove the desired bound, we introduce the following proposition.

**Proposition 3.1.** *Let $\lambda \in (0, 1)$ and $n, m, d \in \mathbb{N}$ with $d$ be even and $m, d \leq n^\lambda$. Let $\nu \in \mathbb{R}_+$ and $A$ be a distribution on $\mathbb{R}^m$ such that for any polynomial $f : \mathbb{R}^m \to \mathbb{R}$ of degree at most $d$ and $\mathbf{E}_{\mathbf{x}\sim\mathcal{N}_m}[f(\mathbf{x})^2] = 1$,*

$$|\mathbf{E}_{\mathbf{x}\sim A}[f(\mathbf{x})] - \mathbf{E}_{\mathbf{x}\sim\mathcal{N}_m}[f(\mathbf{x})]| \leq \nu .$$

*Let $0 < c < (1 - \lambda)/8$ and $n$ is at least a sufficiently large constant depending on $c$, then, for any function $f : \mathbb{R}^n \to [-1, 1]$, it holds*

$$\Pr_{\mathbf{V} \sim U(\mathbf{O}_{n,m})} \left[ \left| \mathbf{E}_{\mathbf{x} \sim \mathbf{P}_{\mathbf{V}}^A} [f(\mathbf{x})] - \mathbf{E}_{\mathbf{x} \sim \mathcal{N}_n} [f(\mathbf{x})] \right| \geq \left( \frac{\Gamma(\frac{d+m}{2})}{\Gamma(\frac{m}{2})} \right) n^{-\left(\frac{1-\lambda}{8} - c\right)d} + (1 + o(1))\nu \right] \leq 2^{-n^{\Omega(c)}} .$$

Assuming Proposition 3.1, the proof of our main theorem is quite simple.

*Proof for Theorem 1.5.* Suppose there is an SQ algorithm $\mathcal{A}$ using $q < 2^{n^{\Omega(c)}}$ many queries of accuracy $\tau \geq \left( \frac{\Gamma(d/2+m/2)}{\Gamma(m/2)} \right) n^{-((1-\lambda)/8-c)d} + (1 + o(1))\nu$ and succeeds with at least $2/3$ probability. We prove by contradiction that such an $\mathcal{A}$ cannot exist. Suppose the input distribution is $\mathcal{N}_n$, and the SQ oracle always answers $\mathbf{E}_{\mathbf{x} \sim \mathcal{N}_n}[f(\mathbf{x})]$ for any query $f$. Then the assumption on $\mathcal{A}$ implies that it answers "null hypothesis" with probability $\alpha > 2/3$. Now consider the case that the input distribution is $\mathbf{P}_{\mathbf{V}}^A$ and $\mathbf{V} \sim U(\mathbf{O}_{n,m})$. Suppose the SQ oracle still always answers $\mathbf{E}_{\mathbf{x} \sim \mathcal{N}_n}[f(\mathbf{x})]$ for any query $f$. Let $f_1, \cdots, f_q$ be the queries the algorithm asks, where $q = 2^{n^{\Omega(c)}}$. By Proposition 3.1 and a union bound, we have

$$\Pr_{\mathbf{V} \sim U(\mathbf{O}_{n,m})}[\exists i \in [q], \ |\mathbf{E}_{\mathbf{x} \sim \mathbf{P}_{\mathbf{V}}^A}[f_i(\mathbf{x})] - \mathbf{E}_{\mathbf{x} \sim \mathcal{N}_n}[f_i(\mathbf{x})]| \geq \tau] = o(1) .$$

Therefore, with probability $1 - o(1)$, the answers given by the oracle are valid. From our assumption on $\mathcal{A}$, the algorithm needs to answer "alternative hypothesis" with probability at least $\frac{2}{3}(1 - o(1))$. But since the oracle always answers $\mathbf{E}_{\mathbf{x} \sim \mathcal{N}_n}[f(\mathbf{x})]$ (which is the same as the above discussed null hypothesis case), we know the algorithm will return "null hypothesis" with probability $\alpha > 2/3$. This gives a contradiction and completes the proof. $\square$

The rest of this section is devoted to the proof of Proposition 3.1.

## 3.1 Fourier Analysis using Hermite Polynomials

The main idea of Proposition 3.1 is to analyze $\mathbf{E}_{\mathbf{x} \sim \mathbf{P}_{\mathbf{V}}^A}[f(\mathbf{x})]$ through Fourier analysis using Hermite polynomials. Before we do the analysis, we will first truncate the distribution $A$ inside $\mathbb{B}^m(B)$ which is the $\ell_2$-norm unit ball in $m$-dimension with radius $B$ for some $B \in \mathbb{R}_+$ to be specified. Namely, we will consider the truncated distribution $A'$ defined as the distribution of $\mathbf{x} \sim A$ conditioned on $\mathbf{x} \in \mathbb{B}^m(B)$. The following lemma shows that given any $m$-dimensional distribution $A$ that approximately matches the first $d$ moment tensor with the Gaussian, the truncated distribution $A'$ is close to $A$ in both the total variation distance and the first $d$ moment tensors.

**Lemma 3.2.** *Let $n, m, d \in \mathbb{N}$ with $d$ be even. Let $A$ be a distribution on $\mathbb{R}^m$ such that for any polynomial $f$ of degree at most $d$ and $\mathbf{E}_{\mathbf{x} \sim \mathcal{N}_m}[f(\mathbf{x})^2] = 1$,*

$$|\mathbf{E}_{\mathbf{x} \sim A}[f(\mathbf{x})] - \mathbf{E}_{\mathbf{x} \sim \mathcal{N}_m}[f(\mathbf{x})]| \leq \nu \leq 2 .$$

*Let $B \in \mathbb{R}_+$ such that $B^d \geq c_1 \left( 2^{d/2} \sqrt{\frac{\Gamma(d+m/2)}{\Gamma(m/2)}} \right)$ where $c_1$ is at least a sufficiently large universal constant and let $A'$ be the truncated distribution defined as the distribution of $\mathbf{x} \sim A$ conditioned on $\mathbf{x} \in \mathbb{B}^m(B)$. Then $d_{\mathrm{TV}}(A, A') = O\left( 2^{d/2} \sqrt{\frac{\Gamma(d+m/2)}{\Gamma(m/2)}} \right) B^{-d}$, Furthermore, any $k \in \mathbb{N}$ and $k < d$,*

$$\|\mathbf{E}_{\mathbf{x} \sim A'}[\mathbf{H}_k(\mathbf{x})] - \mathbf{E}_{\mathbf{x} \sim \mathcal{N}_m}[\mathbf{H}_k(\mathbf{x})]\|_2 = 2^{O(k)} \left( 2^{d/2} \sqrt{\frac{\Gamma(d+m/2)}{\Gamma(m/2)}} \right) B^{-(d-k)} + \nu .$$

The proof of Lemma 3.2 is deferred to Appendix B.

Since $d_{\mathrm{TV}}(A, A') \leq O\left( 2^{d/2} \sqrt{\frac{\Gamma(d+m/2)}{\Gamma(m/2)}} \right) B^{-d}$ and $f$ is bounded in $[-1, 1]$, it follows that

$$|\mathbf{E}_{\mathbf{x} \sim \mathbf{P}_{\mathbf{V}}^{A'}}[f(\mathbf{x})] - \mathbf{E}_{\mathbf{x} \sim \mathbf{P}_{\mathbf{V}}^A}[f(\mathbf{x})]| \leq 2d_{\mathrm{TV}}(\mathbf{P}_{\mathbf{V}}^{A'}, \mathbf{P}_{\mathbf{V}}^{A'}) = 2d_{\mathrm{TV}}(A, A')$$

$$= O\left( \left( 2^{d/2} \sqrt{\frac{\Gamma(d+m/2)}{\Gamma(m/2)}} \right) B^{-d} \right) .$$

Therefore, if suffices for us to analyze $\mathbf{E}_{\mathbf{x} \sim \mathbf{P}_{\mathbf{V}}^{A'}}[f(\mathbf{x})]$ instead of $\mathbf{E}_{\mathbf{x} \sim \mathbf{P}_{\mathbf{V}}^{A}}[f(\mathbf{x})]$. Furthermore, the property that $A'$ is bounded inside $\mathbb{B}^m(B)$ will be convenient in the Fourier analysis later. We introduce the following lemma which decomposes $\mathbf{E}_{\mathbf{x} \sim \mathbf{P}_{\mathbf{V}}^{A'}}[f(\mathbf{x})]$ using Hermite analysis. The proof is deferred to Appendix B.

**Lemma 3.3** (Fourier decomposition Lemma). *Let $A'$ be any distribution supported on $\mathbb{R}^m$, $\mathbf{V} \in \mathbb{R}^{n \times m}$ and $\mathbf{V}^\mathsf{T}\mathbf{V} = \mathbf{I}_m$. Then for any $\ell \in \mathbb{N}$, $\mathbf{E}_{\mathbf{x} \sim \mathbf{P}_{\mathbf{V}}^{A'}}[f(\mathbf{x})] = \sum_{k=0}^{\ell} \langle \mathbf{V}^{\otimes k} \mathbf{A}_k, \mathbf{T}_k \rangle + \mathbf{E}_{\mathbf{x} \sim \mathbf{P}_{\mathbf{V}}^{A'}}[f^{>\ell}(\mathbf{x})]$, where $\mathbf{A}_k = \mathbf{E}_{\mathbf{x} \sim A'}[\mathbf{H}_k(\mathbf{x})]$ and $\mathbf{T}_k = \mathbf{E}_{\mathbf{x} \sim \mathcal{N}_n}[f(\mathbf{x})\mathbf{H}_k(\mathbf{x})]$.*

**Remark 3.4.** Ideally, in Lemma 3.3, we would like to have $\mathbf{E}_{\mathbf{x} \sim \mathbf{P}_{\mathbf{V}}^{A'}}[f(\mathbf{x})] = \sum_{k=0}^{\infty} \langle \mathbf{V}^{\otimes k} \mathbf{A}_k, \mathbf{T}_k \rangle$. However, since we do not assume that $\chi^2(A', \mathcal{N}_m) < \infty$, this convergence may not hold.

Recall that our goal is to show that $|\mathbf{E}_{\mathbf{x} \sim \mathbf{P}_{\mathbf{V}}^{A}}[f(\mathbf{x})] - \mathbf{E}_{\mathbf{x} \sim \mathcal{N}_n}[f(\mathbf{x})]|$ is small with high probability. Observe that $\mathbf{E}_{\mathbf{x} \sim \mathcal{N}_n}[f(\mathbf{x})] = \mathbf{T}_0$ which is the first term in the summation of $\sum_{k=0}^{\ell} \langle \mathbf{V}^{\otimes k} \mathbf{A}_k, \mathbf{T}_k \rangle$ (since $\mathbf{A}_0 = 1$). Therefore, given Lemma 3.3, it suffices to show that $|\sum_{k=1}^{\ell} \langle \mathbf{V}^{\otimes k} \mathbf{A}_k, \mathbf{T}_k \rangle|$ and $|\mathbf{E}_{\mathbf{x} \sim \mathbf{P}_{\mathbf{V}}^{A'}}[f^{\geq \ell}(\mathbf{x})]|$ are both small with high probability. We ignore the $|\mathbf{E}_{\mathbf{x} \sim \mathbf{P}_{\mathbf{V}}^{A'}}[f^{\geq \ell}(\mathbf{x})]|$ part for now as this is mostly a technical issue. To bound $|\sum_{k=1}^{\ell} \langle \mathbf{V}^{\otimes k} \mathbf{A}_k, \mathbf{T}_k \rangle|$, it suffices to analyze $\sum_{k=1}^{\ell} |\langle \mathbf{V}^{\otimes k} \mathbf{A}_k, \mathbf{T}_k \rangle|$ by looking at each term $|\langle \mathbf{V}^{\otimes k} \mathbf{A}_k, \mathbf{T}_k \rangle| = |\langle \mathbf{A}_k, (\mathbf{V}^\mathsf{T})^{\otimes k} \mathbf{T}_k \rangle| \leq \|\mathbf{A}_k\|_2 \|(\mathbf{V}^\mathsf{T})^{\otimes k} \mathbf{T}_k\|_2$. To show that the summation is small, given we need to prove that (with high probability):

1. $\|\mathbf{A}_k\|_2$ does not grow too fast w.r.t $k$;

2. $\|(\mathbf{V}^\mathsf{T})^{\otimes k} \mathbf{T}_k\|_2$ decays very fast w.r.t $k$ (is small with high probability w.r.t the randomness of $\mathbf{V}$).

$\|\mathbf{A}_k\|_2$ **does not grow too fast:** We will use slightly different arguments depending on the size of $k$. We consider three cases : $k < m$, $m \leq k \leq n^{(1-\lambda)/4}$, and $k \geq n^{(1-\lambda)/4}$ (the value in the exponent will deviate by a small quantity to make the proof go through). For $k < m$, $\|\mathbf{A}_k\|_2$ grows slowly by the approximate moment-matching property of $A'$. For $m \leq k \leq n^{(1-\lambda)/4}$, we require the following fact:

**Fact 3.5.** *Let $\mathbf{H}_k$ be the $k$-th Hermite tensor for $m$-dimension. Suppose $\|\mathbf{x}\|_2^2 \geq m$, then $\|\mathbf{H}_k(\mathbf{x})\|_2 = 2^{O(k)} \max(\|\mathbf{x}\|_2^k, 1)$.*

We provide the proof of Fact 3.5 in Appendix B. For $k > n^{(1-\lambda)/4}$, we can show that $\|\mathbf{A}_k\|_2$ does not grow too fast by the following asymptotic bound on Hermite tensors.

**Fact 3.6.** *Let $\mathbf{H}_k$ be the $k$-th Hermite tensor for $m$-dimension, then $\|\mathbf{H}_k(\mathbf{x})\|_2 \leq 2^{O(m)} \binom{k+m-1}{m-1}^{1/2} \exp(\|\mathbf{x}\|_2^2/4)$.*

We provide the proof of Fact 3.6 in Appendix B.

$\|(\mathbf{V}^\mathsf{T})^{\otimes k} \mathbf{T}_k\|_2$ **decays very fast:** We show that $|\langle \mathbf{V}^{\otimes k}, \mathbf{T}_k \rangle|$ is small with high probability by bounding its $a$-th moment for some even $a$. Notice that since $\|\mathbf{H}_k\|_2 \leq \|f\|_2 \leq 1$, we can then combine it with the following lemma. We defer its proof to Appendix B.

**Lemma 3.7.** *Let $k \in \mathbb{Z}_+$, $a \in \mathbb{Z}_+$ be even, $\mathbf{T} \in \mathbb{R}^{n^k}$ and $m \in \mathbb{Z}_+$ satisfy $m < n$. Then, we have*

$$\mathbf{E}_{\mathbf{V} \sim U(\mathbf{O}_{n,m})}[\|(\mathbf{V}^\mathsf{T})^{\otimes k} \mathbf{T}\|_2^a] \leq \mathbf{E}_{\mathbf{V} \sim U(\mathbf{O}_{n,m})}\left[\|\mathbf{V}^\mathsf{T}\mathbf{u}\|_2^{ak/2}\right] \|\mathbf{T}\|_2^a .$$

Roughly speaking, this is just the $ak/2$-th moment of the correlation between a random subspace and a random direction, which can be upper bounded by the following lemma and corollary (see Appendix B for the proofs).

**Lemma 3.8.** *For any even $k \in \mathbb{N}$, and $\mathbf{u} \in \mathbb{S}^{n-1}$, $\mathbf{E}_{\mathbf{V}^\mathsf{T} \sim U(\mathbf{O}_{n,m})}[\|\mathbf{V}\mathbf{u}\|_2^k] = \Theta\left(\frac{\Gamma\left(\frac{k+m}{2}\right)\Gamma\left(\frac{n}{2}\right)}{\Gamma\left(\frac{k+n}{2}\right)\Gamma\left(\frac{m}{2}\right)}\right).$*

**Corollary 3.9.** *Let $c \in (0,1)$ and $m \leq n^c$. Let $k \in \mathbb{N}$ be even. We have that*

$$\mathbf{E}_{\mathbf{V} \sim U(\mathbf{O}_{n,m})}[\|\mathbf{V}^\mathsf{T}\mathbf{u}\|_2^k] = \begin{cases} O(2^{k/2} n^{-(1-c)k/2}) & k \leq n^c\,, \\ \exp(-\Omega(n^c)) O\left(\left(\frac{n^c + n}{k+n}\right)^{(n-m)/2}\right) & k \geq n^c\,. \end{cases}$$

To combine the above results and give the high probability upper bound on $\sum_{k=1}^{\ell} |\langle \mathbf{A}_k, (\mathbf{V}^\mathsf{T})^{\otimes k} \mathbf{T}_k \rangle|$, we require the following lemma. The proof is deferred to Appendix B.

**Lemma 3.10.** *Under the conditions of Proposition 3.1, and further assume $d, m \leq n^\lambda / \log n$, $\nu < 2$ and $\left(\frac{\Gamma(d/2+m/2)}{\Gamma(m/2)}\right) n^{-((1-\lambda)/8-c)d} < 2$. Then for any $n$ that is at least a sufficiently small constant depending on $c$, there is an $\alpha' < (1-\lambda)/8$ such that for any $B = n^\alpha$ where $\alpha' < \alpha < (1-\lambda)/8$ the following holds. Let $A'$ be the truncated distribution defined as the distribution of $\mathbf{x} \sim A$ conditioned on $\mathbf{x} \in \mathbb{B}^m(B)$. Then $d_{\mathrm{TV}}(A, A') \leq \left(\frac{\Gamma(d/2+m/2)}{\Gamma(m/2)}\right) n^{-((1-\lambda)/8-c)d}$. Furthermore for any $\ell \in \mathbb{N}$, except with probability at most $2^{-n^{\Omega(c)}}$ w.r.t. $\mathbf{V} \sim U(\mathbf{O}_{n,m})$, $\sum_{k=1}^{\ell} |\langle \mathbf{A}_k, (\mathbf{V}^\mathsf{T})^{\otimes k} \mathbf{T}_k \rangle| = \left(\frac{\Gamma(d/2+m/2)}{\Gamma(m/2)}\right) n^{-((1-\lambda)/8-c)d} + (1 + o(1))\nu$ where $\mathbf{A}_k = \mathbf{E}_{\mathbf{x} \sim A'}[\mathbf{H}_k(\mathbf{x})]$ and $\mathbf{T}_k = \mathbf{E}_{\mathbf{x} \sim \mathcal{N}_n}[f(\mathbf{x})\mathbf{H}_k(\mathbf{x})]$.*

### 3.2 Proof for Proposition 3.1

We are now ready to prove Proposition 3.1 which is the main technical ingredient of our main result. Proposition 3.1 states that $|\mathbf{E}_{\mathbf{x} \sim \mathbf{P}_{\mathbf{V}}^A}[f(\mathbf{x})] - \mathbf{E}_{\mathbf{x} \sim \mathcal{N}_n}[f(\mathbf{x})]|$ is small with high probability. The main idea of the proof is to use Fourier analysis on $\mathbf{E}_{\mathbf{x} \sim \mathbf{P}_{\mathbf{V}}^{A'}}[f(\mathbf{x})]$ as we discussed in the last section, where $A'$ is the the distribution obtained by truncating $A$ inside $\mathbb{B}^m(B)$.

*Proof for Proposition 3.1.* For convenience, we let $\zeta = (1 - \lambda)/8 - c$. We will analyze $\mathbf{E}_{\mathbf{x} \sim A}[\mathbf{H}_k(\mathbf{x})]$ by truncating $A$. Therefore, we will apply Lemma 3.10 here. However, notice Lemma 3.10 additionally assumes $d, m \leq n^\lambda / \log n$, $\nu < 2$ and $\left(\frac{\Gamma(d/2+m/2)}{\Gamma(m/2)}\right) n^{-\zeta d} < 2$. We show that all these three conditions can be assumed true WLOG. If either the second or the third condition is not true, then our lower bound here is trivialized and is always true since $f$ is bounded between $[-1, +1]$. For $d, m \leq n^\lambda / \log n$, consider a $\lambda' > \lambda$ such that $(1 - \lambda')/8 - \zeta = \frac{(1-\lambda)/8-\zeta}{2}$. Then it is easy to see for any sufficiently large $n$ depending on $(1 - \lambda)/8 - \zeta$, we have $d, m \leq n^{\lambda'} / \log n$ and $\zeta \leq (1 - \lambda)/8 - \zeta$. Therefore, we can WLOG apply Lemma 3.10 for $\lambda'$.

Now let $B = n^\alpha$ where $\alpha < (1 - \lambda)/8$ is the constant in Lemma 3.10. Then we consider the truncated distribution $A'$ defined as the distribution of $\mathbf{x} \sim A$ conditioned on $\mathbf{x} \in \mathbb{B}^m(B)$. By Lemma 3.10, we have $d_{\mathrm{TV}}(A, A') \leq \left(\frac{\Gamma(d/2+m/2)}{\Gamma(m/2)}\right) n^{-\zeta d}$. Given that $f$ is bounded between $[-1, 1]$, this implies $|\mathbf{E}_{\mathbf{x} \sim \mathbf{P}_{\mathbf{V}}^A}[f(\mathbf{x})] - \mathbf{E}_{\mathbf{x} \sim \mathbf{P}_{\mathbf{V}}^{A'}}[f(\mathbf{x})]| \leq 2d_{\mathrm{TV}}(\mathbf{P}_{\mathbf{V}}^A, \mathbf{P}_{\mathbf{V}}^{A'}) = 2d_{\mathrm{TV}}(A, A') \leq 2\left(\frac{\Gamma(d/2+m/2)}{\Gamma(m/2)}\right) n^{-\zeta d}$. Thus, it suffices for us to analyze $\mathbf{E}_{\mathbf{x} \sim \mathbf{P}_{\mathbf{V}}^{A'}}[f(\mathbf{x})]$ instead of $\mathbf{E}_{\mathbf{x} \sim \mathbf{P}_{\mathbf{V}}^A}[f(\mathbf{x})]$.

Let $\ell = \ell_f(n) \in \mathbb{N}$ be a function depending only on the query function $f$ and the dimension $n$ ($\ell$ to be specified later). By Lemma 3.3, we have that $\mathbf{E}_{\mathbf{x} \sim \mathbf{P}_{\mathbf{V}}^{A'}}[f(\mathbf{x})] = \sum_{k=0}^{\ell} |\langle \mathbf{A}_k, (\mathbf{V}^\mathsf{T})^{\otimes k} \mathbf{T}_k \rangle| + \mathbf{E}_{\mathbf{x} \sim \mathbf{P}_{\mathbf{V}}^{A'}}[f^{>\ell}(\mathbf{x})]$. Recall that we want to bound $|\mathbf{E}_{\mathbf{x} \sim \mathbf{P}_{\mathbf{V}}^{A'}}[f(\mathbf{x})] - \mathbf{E}_{\mathbf{x} \sim \mathcal{N}_n}[f(\mathbf{x})]|$ with high probability, where we note that $\mathbf{E}_{\mathbf{x} \sim \mathcal{N}_n}[f(\mathbf{x})] = \langle \mathbf{A}_0, \mathbf{T}_0 \rangle$. Therefore, we can write

$$|\mathbf{E}_{\mathbf{x} \sim \mathbf{P}_{\mathbf{V}}^{A'}}[f(\mathbf{x})] - \mathbf{E}_{\mathbf{x} \sim \mathcal{N}_n}[f(\mathbf{x})]| \leq \left|\sum_{k=1}^{\ell} \langle \mathbf{A}_k, (\mathbf{V}^\mathsf{T})^{\otimes k} \mathbf{T}_k \rangle\right| + |\mathbf{E}_{\mathbf{x} \sim \mathbf{P}_{\mathbf{V}}^{A'}}[f^{>\ell}(\mathbf{x})]|\,.$$

For the first term, by Lemma 3.10, we have that $|\sum_{k=1}^{\ell} \langle \mathbf{A}_k, (\mathbf{V}^\mathsf{T})^{\otimes k} \mathbf{T}_k \rangle| = \left(\frac{\Gamma(d/2+m/2)}{\Gamma(m/2)}\right) n^{-\zeta d} + (1 + o(1))\nu$, except with probability $2^{-n^{\Omega(c)}}$.

It now remains for us to show that $|\mathbf{E}_{\mathbf{x} \sim \mathbf{P}_{\mathbf{V}}^{A'}}[f^{>\ell}(\mathbf{x})]|$ is also small with high probability. Consider the distribution $D = \mathbf{E}_{\mathbf{v} \sim U(\mathbf{O}_{n,m})}[\mathbf{P}_{\mathbf{V}}^{A'}]$. The following lemma shows that $D$ is continuous

and $\chi^2(D, \mathcal{N}_n)$ is at most a constant only depending on $n$ (independent of the choice of the one dimensional distribution $A$).

**Lemma 3.11.** *Let $A$ be any distribution supported on $\mathbb{B}^m(n)$ for $n \in \mathbb{N}$ which is at least a sufficiently large universal constant. Let $D = \mathbf{E}_{\mathbf{V} \sim U(\mathbf{O}_{n,m})}[\mathbf{P}_{\mathbf{V}}^{A'}]$. Then, $D$ is a continuous distribution and $\chi^2(D, \mathcal{N}_n) = O_n(1)$.*

Roughly speaking, the proof of the lemma follows by noting that the average over $\mathbf{V}$ of $\mathbf{P}_{\mathbf{V}}^A$ is spherically symmetric. We defer its proof to Appendix B.2. Using this lemma, we can write

$$\mathbf{E}_{\mathbf{V} \sim U(\mathbf{O}_{n,m})}\left[\left|\mathbf{E}_{\mathbf{x} \sim \mathbf{P}_{\mathbf{V}}^{A'}}[f^{>\ell}(\mathbf{x})]\right|\right] \leq \mathbf{E}_{\mathbf{V} \sim U(\mathbf{O}_{n,m})}\left[\mathbf{E}_{\mathbf{x} \sim \mathbf{P}_{\mathbf{V}}^{A'}}\left[|f^{>\ell}(\mathbf{x})|\right]\right] = \mathbf{E}_{\mathbf{x} \sim D}\left[|f^{>\ell}(\mathbf{x})|\right]$$

$$= \mathbf{E}_{\mathbf{x} \sim \mathcal{N}_n}\left[\frac{D(\mathbf{x})}{\mathcal{N}_n(\mathbf{x})} |f^{>\ell}(\mathbf{x})|\right] \leq \sqrt{1 + \chi^2(D, \mathcal{N}_n)} \, \|f^{>\ell}\|_2,$$

where the first equality holds due to Fubini's theorem. Furthermore, since $\chi^2(D, \mathcal{N}_n) = O_n(1)$, there is a function $\delta : \mathbb{R} \to \mathbb{R}$ such that $1 + \chi^2(D, \mathcal{N}_n) \leq \delta(n)$. Therefore, we have that

$$\mathbf{E}_{\mathbf{V} \sim U(\mathbf{O}_{n,m})}\left[\left|\mathbf{E}_{\mathbf{x} \sim \mathbf{P}_{\mathbf{V}}^{A'}}[f^{>\ell}(\mathbf{x})]\right|\right] \leq \sqrt{1 + \chi^2(D, \mathcal{N}_n)} \, \|f^{>\ell}\|_2 \leq \delta(n)\|f^{>\ell}\|_2 \, .$$

We can take $\ell = \ell_f(n)$ ($\ell$ only depends on the query function $f$ and dimension $n$) to be a sufficiently large function such that $\|f^{>\ell}\|_2 \leq \left(\frac{e^{-n}}{\delta(n)}\right) \left(\frac{\Gamma(d/2 + m/2)}{\Gamma(m/2)}\right) n^{-\zeta d}$. Then we get

$$\mathbf{E}_{\mathbf{V} \sim U(\mathbf{O}_{n,m})}\left[\left|\mathbf{E}_{\mathbf{x} \sim \mathbf{P}_{\mathbf{V}}^{A'}}[f^{>\ell}(\mathbf{x})]\right|\right] \leq \delta(n)\|f^{>\ell}\|_2 \leq e^{-n}\left(\frac{\Gamma(d/2 + m/2)}{\Gamma(m/2)}\right) n^{-\zeta d} \, .$$

This gives the tail bound $\mathbf{Pr}_{\mathbf{V} \sim U(\mathbf{O}_{n,m})}\left[\left|\mathbf{E}_{\mathbf{x} \sim \mathbf{P}_{\mathbf{V}}^{A'}}[f^{>\ell}(\mathbf{x})]\right| \geq \left(\frac{\Gamma(d/2 + m/2)}{\Gamma(m/2)}\right) n^{-\zeta d}\right] \leq e^{-n}$.

Using the above upper bounds, we have

$$\left|\mathbf{E}_{\mathbf{x} \sim \mathbf{P}_{\mathbf{V}}^{A'}}[f(\mathbf{x})] - \mathbf{E}_{\mathbf{x} \sim \mathcal{N}_n}[f(\mathbf{x})]\right| \leq \left|\sum_{k=1}^{\ell}\langle \mathbf{A}_k, (\mathbf{V}^{\mathsf{T}})^{\otimes k}\mathbf{T}_k\rangle\right| + \left|\mathbf{E}_{\mathbf{x} \sim \mathbf{P}_{\mathbf{V}}^{A'}}[f^{>\ell}(\mathbf{x})]\right|$$

$$= 2\left(\frac{\Gamma(d/2 + m/2)}{\Gamma(m/2)}\right) n^{-\zeta d} + (1 + o(1))\nu \, ,$$

except with probability $2^{-n^{\Omega(1)}}$. As we have argued at the beginning of the proof,

$$\left|\mathbf{E}_{\mathbf{x} \sim \mathbf{P}_{\mathbf{V}}^{A}}[f(\mathbf{x})] - \mathbf{E}_{\mathbf{x} \sim \mathbf{P}_{\mathbf{V}}^{A'}}[f(\mathbf{x})]\right| \leq 2\left(\frac{\Gamma(d/2 + m/2)}{\Gamma(m/2)}\right) n^{-\zeta d} \, .$$

Therefore,

$$\left|\mathbf{E}_{\mathbf{x} \sim \mathbf{P}_{\mathbf{V}}^{A}}[f(\mathbf{x})] - \mathbf{E}_{\mathbf{x} \sim \mathcal{N}_n}[f(\mathbf{x})]\right| \leq 3\left(\frac{\Gamma(d/2 + m/2)}{\Gamma(m/2)}\right) n^{-\zeta d} + (1 + o(1))\nu \, ,$$

except with probability $2^{-n^{\Omega(1)}} < 2^{-n^{\Omega(c)}}$ given $c = O(1)$.

In the end, notice that the above argument is still true if we take $\zeta' > \zeta$ such that $(1-\lambda)/8 - \zeta' = \frac{(1-\lambda)/8 - \zeta}{2}$. Using the above argument for $\zeta'$ and given $n$ is a sufficiently large constant depending $(1-\lambda)/8 - \zeta = 2((1-\lambda)/8 - \zeta')$, we get

$$\left|\mathbf{E}_{\mathbf{x} \sim \mathbf{P}_{\mathbf{V}}^{A}}[f(\mathbf{x})] - \mathbf{E}_{\mathbf{x} \sim \mathcal{N}_n}[f(\mathbf{x})]\right| \leq \left(\frac{\Gamma(d/2 + m/2)}{\Gamma(m/2)}\right) n^{-\zeta d} + (1 + o(1))\nu \, ,$$

except with probability $2^{-n^{\Omega((1-\lambda)/8 - \zeta')}} = 2^{-n^{\Omega(c)}}$. Replacing $\zeta$ with $(1-\lambda)/8 - c$ completes the proof.

$\square$

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
