## Supplementary Material

## A   Omitted Proofs from Section 2

### A.1   Proof of Claim 2.3

*Proof.* The proof of Claim 2.3 is obtained via the following calculation, using the definition of Hermite tensor (Definition 2.2). We will use $i, j$ for indexes in $[d]$.

$$
\begin{aligned}
&\sqrt{k!}\mathbf{H}_k(\mathbf{Bx})_{i_1,\ldots,i_k} \\
=& \sum_{\substack{\text{Partitions } P \text{ of } [k] \\ \text{into sets of size 1 and 2}}} \bigotimes_{\{a,b\}\in P} (-\mathbf{I}_{i_a,i_b}) \bigotimes_{\{c\}\in P} (\mathbf{Bx})_{i_c} \\
=& \sum_{\substack{\text{Partitions } P \text{ of } [k] \\ \text{into sets of size 1 and 2}}} \bigotimes_{\{a,b\}\in P} (-(\mathbf{BIB}^T)_{i_a,i_b}) \bigotimes_{\{c\}\in P} (\mathbf{Bx})_{i_c} \\
=& \sum_{\substack{\text{Partitions } P \text{ of } [k] \\ \text{into sets of size 1 and 2}}} \bigotimes_{\{a,b\}\in P} \left( -\left( \sum_{j_a,j_b=1}^{n} \mathbf{B}_{i_a,j_a}\mathbf{I}_{j_a,j_b}\mathbf{B}^\intercal_{j_b,i_b} \right) \right) \bigotimes_{\{c\}\in P} \left( \sum_{j_c=1}^{n} \mathbf{B}_{i_c,j_c}\mathbf{x}_{j_c} \right) \\
=& \sum_{\substack{\text{Partitions } P \text{ of } [k] \\ \text{into sets of size 1 and 2}}} \sum_{j_1,\ldots,j_k=1}^{n} \bigotimes_{\{a,b\}\in P} \left( \mathbf{B}_{i_a,j_a}(-\mathbf{I}_{j_a,j_b})\mathbf{B}^\intercal_{j_b,i_b} \right) \bigotimes_{\{c\}\in P} (\mathbf{B}_{i_c,j_c}\mathbf{x}_{j_c}) \\
=& \sum_{\substack{\text{Partitions } P \text{ of } [k] \\ \text{into sets of size 1 and 2}}} \sum_{j_1,\ldots,j_k=1}^{n} \bigotimes_{l\in[k]} \mathbf{B}_{i_l,j_l} \bigotimes_{\{a,b\}\in P} (-\mathbf{I}_{j_a,j_b}) \bigotimes_{\{c\}\in P} \mathbf{x}_{j_c} \\
=& \sum_{j_1,\ldots,j_k=1}^{n} \bigotimes_{l\in[k]} \mathbf{B}_{i_l,j_l} \sum_{\substack{\text{Partitions } P \text{ of } [k] \\ \text{into sets of size 1 and 2}}} \bigotimes_{\{a,b\}\in P} (-\mathbf{I}_{j_a,j_b}) \bigotimes_{\{c\}\in P} \mathbf{x}_{j_c} \\
=& \sum_{j_1,\ldots,j_k=1}^{n} \bigotimes_{l\in[k]} \mathbf{B}_{i_l,j_l} \sqrt{k!}\mathbf{H}_k(\mathbf{x})_{j_1,\ldots,j_k} \\
=& \sqrt{k!} \sum_{j_1,\ldots,j_k=1}^{n} (\mathbf{B}^{\otimes k})_{i_1,\ldots,i_k,j_1,\ldots,j_k} \mathbf{H}_k(\mathbf{x})_{j_1,\cdots,j_k},
\end{aligned}
$$

where the fourth and fifth equalities follow from the fact that $P$ is a partition of $[k]$, so changing the order of summation and multiplication gives exactly $\sum_{j_1,\ldots,j_k=1}^{n}$ and $\bigotimes_{l\in[k]} \mathbf{B}_{i_l,j_l}$. The seventh equality follows from the definition of the Hermite tensor. The above is equivalent to $\mathbf{H}_k(\mathbf{Bx}) = \mathbf{B}^{\otimes k}\mathbf{H}_k(\mathbf{x})$. This completes the proof. $\square$

## B   Omitted Proofs from Section 3

### B.1   Omitted Proofs from Section 3.1

#### B.1.1   Proof of Lemma 3.2

*Proof.* We construct the truncated distribution $A'$ as follows. We first sample $\mathbf{x} \sim A$, then we reject $\mathbf{x}$ unless $\|\mathbf{x}\|_2 \leq B$. Let $A'$ be the distribution of the samples we get from this process.

First notice that to bound the total variation distance,

$$
\begin{aligned}
\mathbf{E}_{\mathbf{x}\sim A}[\|\mathbf{x}\|_2^d] \leq & \mathbf{E}_{\mathbf{x}\sim\mathcal{N}_m}[\|\mathbf{x}\|_2^d] + \nu\mathbf{E}_{\mathbf{x}\sim\mathcal{N}_m}[\|\mathbf{x}\|_2^{2d}]^{1/2} \\
= & \mathbf{E}_{t\sim\chi^2(m)}[t^{d/2}] + \nu\mathbf{E}_{t\sim\chi^2(m)}[t^d]^{1/2} \\
= & 2^{d/2}\frac{\Gamma((d+m)/2)}{\Gamma(m/2)} + 2^{d/2}\sqrt{\frac{\Gamma((2d+m)/2)}{\Gamma(m/2)}}\nu \\
\leq & c_2\left(2^{d/2}\sqrt{\frac{\Gamma(d+m/2)}{\Gamma(m/2)}}\right),
\end{aligned}
$$

where $c_2$ is a universal constant. Using Markov's inequality and union bound, we have

$$
\mathbf{Pr}_{\mathbf{x}\sim A}[\mathbf{x}\notin\mathbb{B}^m(B)] \leq c_2\left(2^{d/2}\sqrt{\frac{\Gamma(d+m/2)}{\Gamma(m/2)}}\right)B^{-d}.
$$

By the definition of $A'$, we have that $d_{\mathrm{TV}}(A,A') = c_2\left(2^{d/2}\sqrt{\frac{\Gamma(d+m/2)}{\Gamma(m/2)}}\right)B^{-d}$.

Then it only remains to verify $\|\mathbf{E}_{\mathbf{x}\sim A'}[\mathbf{H}_k(\mathbf{x})] - \mathbf{E}_{\mathbf{x}\sim\mathcal{N}_m}[\mathbf{H}_k(\mathbf{x})]\|_2$ for any $k < d$. It is immediate that $\|\mathbf{E}_{\mathbf{x}\sim A'}[\mathbf{H}_k(\mathbf{x})] - \mathbf{E}_{\mathbf{x}\sim\mathcal{N}_m}[\mathbf{H}_k(\mathbf{x})]\|_2 = 0$ for $k = 0$. Therefore, we only consider $1 \leq k < d$. We first look at $\|\mathbf{E}_{\mathbf{x}\sim A}[\mathbf{H}_k(\mathbf{x})] - \mathbf{E}_{\mathbf{x}\sim\mathcal{N}_m}[\mathbf{H}_k(\mathbf{x})]\|_2$. Suppose $\|\mathbf{E}_{\mathbf{x}\sim A}[\mathbf{H}_k(\mathbf{x})] - \mathbf{E}_{\mathbf{x}\sim\mathcal{N}_m}[\mathbf{H}_k(\mathbf{x})]\|_2 > \nu$, then it is easy to see that the polynomial

$$
f = \left\langle \frac{\mathbf{E}_{\mathbf{x}\sim A}[\mathbf{H}_k(\mathbf{x})] - \mathbf{E}_{\mathbf{x}\sim\mathcal{N}_m}[\mathbf{H}_k(\mathbf{x})]}{\|\mathbf{E}_{\mathbf{x}\sim A}[\mathbf{H}_k(\mathbf{x})] - \mathbf{E}_{\mathbf{x}\sim\mathcal{N}_m}[\mathbf{H}_k(\mathbf{x})]\|_2}, \mathbf{H}_k(\mathbf{x}) \right\rangle
$$

satisfies the requirement, that $f$ is at most degree-$d$, $\mathbf{E}_{\mathbf{x}\sim\mathcal{N}_m}[f(\mathbf{x})^2] = 1$ and $|\mathbf{E}_{\mathbf{x}\sim A}[f(\mathbf{x})] - \mathbf{E}_{\mathbf{x}\sim\mathcal{N}_m}[f(\mathbf{x})]| > \nu$. Contradiction, thus $\|\mathbf{E}_{\mathbf{x}\sim A}[\mathbf{H}_k(\mathbf{x})] - \mathbf{E}_{\mathbf{x}\sim\mathcal{N}_m}[\mathbf{H}_k(\mathbf{x})]\|_2 \leq \nu$. Then to bound $\|\mathbf{E}_{\mathbf{x}\sim A'}[\mathbf{H}_k(\mathbf{x})] - \mathbf{E}_{\mathbf{x}\sim A}[\mathbf{H}_k(\mathbf{x})]\|_2$, let $\alpha = \mathbf{Pr}_{\mathbf{x}\sim A}[\mathbf{x}\notin\mathbb{B}^m(B)]$,

$$
\begin{aligned}
& \|\mathbf{E}_{\mathbf{x}\sim A'}[\mathbf{H}_k(\mathbf{x})] - \mathbf{E}_{\mathbf{x}\sim A}[\mathbf{H}_k(\mathbf{x})]\|_2 \\
= & \left\|\frac{1}{1-\alpha}\mathbf{E}_{\mathbf{x}\sim A}[\mathbf{H}_k(\mathbf{x})\mathbf{1}(\mathbf{x}\in\mathbb{B}^m(B))] - \mathbf{E}_{\mathbf{x}\sim A}[\mathbf{H}_k(\mathbf{x})]\right\|_2 \\
= & \left\|\frac{\alpha}{1-\alpha}\mathbf{E}_{\mathbf{x}\sim A}[\mathbf{H}_k(\mathbf{x})] - \frac{1}{1-\alpha}\mathbf{E}_{\mathbf{x}\sim A}[\mathbf{H}_k(\mathbf{x})\mathbf{1}(\mathbf{x}\notin\mathbb{B}^m(B))]\right\|_2 \\
\leq & \frac{1}{1-\alpha}\|\mathbf{E}_{\mathbf{x}\sim A}[\mathbf{H}_k(\mathbf{x})\mathbf{1}(\mathbf{x}\notin\mathbb{B}^m(B))]\|_2 + \frac{\alpha}{1-\alpha}\|\mathbf{E}_{\mathbf{x}\sim A}[\mathbf{H}_k(\mathbf{x})]\|_2 \\
\leq & \frac{1}{1-\alpha}\|\mathbf{E}_{\mathbf{x}\sim A}[\mathbf{H}_k(\mathbf{x})\mathbf{1}(\mathbf{x}\notin\mathbb{B}^m(B))]\|_2 + \frac{\alpha}{1-\alpha}\nu,
\end{aligned}
$$

where the last inequality follows from $\|\mathbf{E}_{\mathbf{x}\sim A}[\mathbf{H}_k(\mathbf{x})] - \mathbf{E}_{\mathbf{x}\sim\mathcal{N}_m}[\mathbf{H}_k(\mathbf{x})]\|_2 \leq \nu$ and $\mathbf{E}_{\mathbf{x}\sim\mathcal{N}_m}[\mathbf{H}_k(\mathbf{x})] = 0$ for any $k > 0$. Since $B^d \geq c_1\left(2^{d/2}\sqrt{\frac{\Gamma(d+m/2)}{\Gamma(m/2)}}\right)$ ($c_1$ is at least a sufficiently large universal constant), we have $\alpha \geq c_2\left(2^{d/2}\sqrt{\frac{\Gamma(d+m/2)}{\Gamma(m/2)}}\right)B^{-d} \geq 1/2$. Also, we have $B^d \geq c_1\left(2^{d/2}\sqrt{\frac{\Gamma(d+m/2)}{\Gamma(m/2)}}\right)$ implies $B^2 \geq m$. Then using Jensen's inequality and Fact 3.5, we

have

$$\|\mathbf{E}_{\mathbf{x}\sim A}[\mathbf{H}_k(\mathbf{x})\mathbf{1}(\mathbf{x}\notin\mathbb{B}^m(B))]\|_2$$

$$\leq\mathbf{E}_{\mathbf{x}\sim A}[\|\mathbf{H}_k(\mathbf{x})\|_2\mathbf{1}(\mathbf{x}\notin\mathbb{B}^m(B))]$$

$$\leq\mathbf{E}_{\mathbf{x}\sim A}[2^{O(k)}\max(1,\|\mathbf{x}\|_2^k)\mathbf{1}(\mathbf{x}\notin\mathbb{B}^m(B))]$$

$$\leq 2^{O(k)}\mathbf{E}_{\mathbf{x}\sim A}[\|\mathbf{x}\|_2^k\mathbf{1}(\mathbf{x}\notin\mathbb{B}^m(B))]$$

$$\leq 2^{O(k)}\int_0^\infty\mathbf{Pr}[\|\mathbf{x}\|_2\geq u\wedge\mathbf{x}\notin\mathbb{B}^m(B)]du^k$$

$$\leq 2^{O(k)}\int_0^\infty\left(2^{d/2}\sqrt{\frac{\Gamma(d+m/2)}{\Gamma(m/2)}}\right)\min(B^{-d},u^{-d})du^k$$

$$\leq 2^{O(k)}\left(2^{d/2}\sqrt{\frac{\Gamma(d+m/2)}{\Gamma(m/2)}}\right)\left(\int_0^B B^{-d}du^k+\int_B^\infty u^{-d}du^k\right)$$

$$\leq 2^{O(k)}\left(2^{d/2}\sqrt{\frac{\Gamma(d+m/2)}{\Gamma(m/2)}}\right)B^{-(d-k)}\;.$$

Plug it back gives

$$\|\mathbf{E}_{\mathbf{x}\sim A'}[\mathbf{H}_k(\mathbf{x})]-\mathbf{E}_{\mathbf{x}\sim A}[\mathbf{H}_k(\mathbf{x})]\|_2\leq 2^{O(k)}\left(2^{d/2}\sqrt{\frac{\Gamma(d+m/2)}{\Gamma(m/2)}}\right)B^{-(d-k)}+\nu\;.$$

This completes the proof. $\square$

### B.1.2   Proof of Lemma 3.3

*Proof.* Noting that $\mathbf{E}_{\mathbf{x}\sim\mathbf{P}_{\mathbf{V}}^{A'}}[f(\mathbf{x})]=\mathbf{E}_{\mathbf{x}\sim\mathbf{P}_{\mathbf{V}}^{A'}}[f^{\leq\ell}(\mathbf{x})]+\mathbf{E}_{\mathbf{x}\sim\mathbf{P}_{\mathbf{V}}^{A'}}[f^{>\ell}(\mathbf{x})]$, we will show that

$$\mathbf{E}_{\mathbf{x}\sim\mathbf{P}_{\mathbf{V}}^{A'}}[f^{\leq\ell}(\mathbf{x})]=\left\langle\sum\nolimits_{k=0}^\ell\langle\mathbf{A}_k,\mathbf{H}_k(\langle\mathbf{v}_1,\mathbf{x}\rangle,\ldots,\langle\mathbf{v}_m,\mathbf{x}\rangle)\rangle,f^{\leq\ell}\right\rangle_{\mathcal{N}_n}\;.$$

Notice that $f^{\leq\ell}$ is a polynomial of degree-$\ell$. Let $\mathbf{b}_1,\cdots,\mathbf{b}_n$ be an orthonormal basis, where $\mathbf{b}_1=\mathbf{v}_1,\ldots,\mathbf{b}_m=\mathbf{v}_m$. Then degree-$\ell$ polynomials are spanned by polynomials of the form $\prod_{i=1}^n\langle\mathbf{b}_i,\mathbf{x}\rangle^{q_i}$, where $\sum_{i=1}^n q_i\leq\ell$. Therefore, it suffices to show that for any $\sum_{i=1}^n q_i\leq\ell$,

$$\mathbf{E}_{\mathbf{x}\sim\mathbf{P}_{\mathbf{V}}^{A'}}\left[\prod\nolimits_{i=1}^n\langle\mathbf{b}_i,\mathbf{x}\rangle^{q_i}\right]=\left\langle\sum\nolimits_{k=0}^\ell\langle\mathbf{A}_k,\mathbf{H}_k(\langle\mathbf{v}_1,\mathbf{x}\rangle,\ldots,\langle\mathbf{v}_m,\mathbf{x}\rangle)\rangle,\prod\nolimits_{i=1}^n\langle\mathbf{b}_i,\mathbf{x}\rangle^{q_i}\right\rangle_{\mathcal{N}_n}\;,$$

which is equivalent to show that for any $\sum_{i=1}^m q_i\leq\ell$,

$$\mathbf{E}_{\mathbf{x}\sim\mathbf{P}_{\mathbf{V}}^{A'}}\left[\prod\nolimits_{i=1}^m\langle\mathbf{v}_i,\mathbf{x}\rangle^{q_i}\right]=\left\langle\sum\nolimits_{k=0}^\ell\langle\mathbf{A}_k,\mathbf{H}_k(\langle\mathbf{v}_1,\mathbf{x}\rangle,\ldots,\langle\mathbf{v}_m,\mathbf{x}\rangle)\rangle,\prod\nolimits_{i=1}^m\langle\mathbf{v}_i,\mathbf{x}\rangle^{q_i}\right\rangle_{\mathcal{N}_m}\;,$$

which is the same as for any $q=\sum_{i=1}^m q_i\leq\ell$, $\mathbf{E}_{\mathbf{y}\sim A'}[\mathbf{H}_q(\mathbf{y})]=\sum_{k=0}^\ell\mathbf{E}_{\mathbf{y}\sim\mathcal{N}_m}[\langle\mathbf{A}_k,\mathbf{H}_k(\mathbf{y})\rangle\mathbf{H}_q(\mathbf{y})]$. One can see that the above holds from the definition of $\mathbf{A}_k$ and the orthornomal property of Hermite Tensors. Therefore, by Claim 2.3, we have that

$$\mathbf{E}_{\mathbf{x}\sim\mathbf{P}_{\mathbf{V}}^{A'}}[f^{\leq\ell}(\mathbf{x})]=\left\langle\sum\nolimits_{k=0}^\ell\langle\mathbf{A}_k,\mathbf{H}_k(\langle\mathbf{v}_1,\mathbf{x}\rangle,\ldots,\langle\mathbf{v}_m,\mathbf{x}\rangle)\rangle,f^{\leq\ell}\right\rangle_{\mathcal{N}_n}$$

$$=\left\langle\sum\nolimits_{k=0}^\ell\langle\mathbf{A}_k,\mathbf{H}_k(\mathbf{V}^\mathsf{T}\mathbf{x})\rangle,f^{\leq\ell}\right\rangle_{\mathcal{N}_n}$$

$$=\left\langle\sum\nolimits_{k=0}^\ell\langle\mathbf{V}^{\otimes k}\mathbf{A}_k,\mathbf{H}_k(\mathbf{x})\rangle,f^{\leq\ell}\right\rangle_{\mathcal{N}_n}$$

$$=\left\langle\sum\nolimits_{k=0}^\ell\langle\mathbf{V}^{\otimes k}\mathbf{A}_k,\mathbf{H}_k(\mathbf{x})\rangle,\sum\nolimits_{k=0}^\ell\langle\mathbf{T}_k,\mathbf{H}_k(\mathbf{x})\rangle\right\rangle_{\mathcal{N}_n}$$

$$=\sum\nolimits_{k=0}^\ell\langle\mathbf{V}^{\otimes k}\mathbf{A}_k,\mathbf{T}_k\rangle\;,$$

where the last equality uses the orthonormal property of Hermite tensors. This completes the proof.
$\square$

### B.1.3 Proof of Fact 3.5

*Proof.* For a degree-$k$ tensor $\mathbf{A}$, we use $\mathbf{A}^\pi$ to denote the matrix that $\mathbf{A}^\pi_{i_1,\cdots,i_k} = \mathbf{A}_{\pi(i_1,\cdots,i_k)}$. Notice $\|\mathbf{A}\|_2 = \|\mathbf{A}^\pi\|_2$. Then from the definition of Hermite tensor, we have

$$\mathbf{H}(\mathbf{x}) = \frac{1}{\sqrt{k!}} \sum_{t=0}^{\lfloor k/2 \rfloor} \sum_{\text{Permutation } \pi \text{ of } [k]} \frac{1}{2^t t!(k-2t)!} \left( (-\mathbf{I})^{\otimes t} \mathbf{x}^{\otimes(k-2t)} \right)^\pi .$$

This implies,

$$
\begin{aligned}
&\|\mathbf{H}_k(\mathbf{x})]\|_2 \\
&= \left\| \frac{1}{\sqrt{k!}} \sum_{t=0}^{\lfloor k/2 \rfloor} \sum_{\text{Permutation } \pi \text{ of } [k]} \frac{1}{2^t t!(k-2t)!} \left( (-\mathbf{I})^{\otimes t} \mathbf{x}^{\otimes(k-2t)} \right)^\pi \right\|_2 \\
&\leq \sum_{t=1}^{\lfloor k/2 \rfloor} \frac{\sqrt{k!}}{2^t t!(k-2t)!} \max(\|\mathbf{I}^{\otimes t}\|_2 \|\mathbf{x}\|_2^{k-2t}, 1) \\
&= \sum_{t=1}^{\lfloor k/2 \rfloor} \frac{\sqrt{k!}}{2^t t!(k-2t)!} \max(m^t \|\mathbf{x}\|_2^{k-2t}, 1) \\
&= \max(\|\mathbf{x}\|_2^k, 1) \sum_{t=1}^{\lfloor k/2 \rfloor} \frac{\sqrt{k!}}{2^t t!(k-2t)!} .
\end{aligned}
$$

One can see the denominator is minimized when $t = k/2 - O(\sqrt{k})$. Then it follows that the sum is at most $2^{O(k)} \max(\|\mathbf{x}\|_2^k, 1)$. $\qquad \square$

### B.1.4 Proof of Fact 3.6

*Proof.* We will need the following fact for the proof.

**Fact B.1** ([Kra04]). *Let $h_k$ be the $k$-th normalized Hermite polynomial. Then* $\max_{t \in \mathbb{R}} h_k^2(t) e^{-t^2/2} = O(k^{-1/6})$.

First note that using Lemma 2.3, we have for any orthogonal $\mathbf{B} \in \mathbb{R}^{m \times m}$,

$$\|\mathbf{H}_k(\mathbf{x})\|_2 = \|\mathbf{B}^{\otimes k} \mathbf{H}_k(\mathbf{x})\|_2 = \|\mathbf{H}_k(\mathbf{B}\mathbf{x})\|_2 .$$

By taking the appropriate $\mathbf{B}$, we can always have $\mathbf{B}\mathbf{x} = \|\mathbf{x}\|_2 \mathbf{e}_1$. Therefore, wlog, we can assume $\mathbf{x} = t\mathbf{e}_1$ for some $t \in \mathbb{R}$.

Notice that for any entry $H_k(\mathbf{x})_{i_1,\cdots,i_k}$, let $j_\ell$ for $\ell \in 1, \cdots, m$ be the number of times $\ell$ appears in $i_1, \cdots, i_k$. Note that $\sum_\ell j_\ell = k$. To bound the norm of $\mathbf{H}_k(\mathbf{x})$ notice

$$\mathbf{H}_k(\mathbf{x})_{i_1,\cdots,i_k} = \binom{k}{j_1, \cdots, j_m}^{-1/2} \prod_{\ell=1}^m h_{j_\ell}(\mathbf{x}_\ell) .$$

Therefore,

$$
\begin{aligned}
\|\mathbf{H}_k(\mathbf{x})\|_2^2 &= \sum_{i_1,\cdots,i_k} \binom{k}{j_1, \cdots, j_m}^{-1} \left( \prod_{\ell=1}^m h_{j_\ell}(\mathbf{x}_\ell) \right)^2 \\
&= \sum_{j_1,\cdots,j_m \text{ such that } \sum_\ell j_\ell = k} \left( \prod_{\ell=1}^m h_{j_\ell}(\mathbf{x}_\ell) \right)^2 \\
&\leq \sum_{j_1,\cdots,j_m \text{ such that } \sum_\ell j_\ell = k} \prod_{\ell=1}^m O(\exp(\mathbf{x}_\ell^2/2)) \\
&\leq 2^{O(m)} \binom{k+m-1}{m-1} \exp(\|\mathbf{x}\|_2^2/2) ,
\end{aligned}
$$

where the first inequality follows from Fact B.1. Then we can take square root on both sides which gives what we want. $\qquad \square$

### B.1.5 Proof of Lemma 3.7

*Proof.* Notice that

$$
\begin{aligned}
\mathbf{E}_{\mathbf{V}\sim U(\mathbf{O}_{n,m})}[\|(\mathbf{V}^{\mathsf{T}})^{\otimes k}\mathbf{T}\|_2^a] &= \mathbf{E}_{\mathbf{V}\sim U(\mathbf{O}_{n,m})}[\|(\mathbf{V}^{\mathsf{T}})^{\otimes ak/2}\mathbf{T}^{a/2}\|_2^2] \\
&= \mathbf{E}_{\mathbf{V}\sim U(\mathbf{O}_{n,m})}[\langle \mathbf{V}^{\otimes ak/2}(\mathbf{V}^{\mathsf{T}})^{\otimes ak/2}, \mathbf{T}^{\otimes a}\rangle] \\
&= \langle \mathbf{E}_{\mathbf{V}\sim U(\mathbf{O}_{n,m})}[\mathbf{V}^{\otimes ak/2}(\mathbf{V}^{\mathsf{T}})^{\otimes ak/2}], \mathbf{T}^{\otimes a}\rangle \\
&\leq \|\mathbf{E}_{\mathbf{V}\sim U(\mathbf{O}_{n,m})}[\mathbf{V}^{\otimes ak/2}(\mathbf{V}^{\mathsf{T}})^{\otimes ak/2}]\|_2 \|\mathbf{T}\|_2^a .
\end{aligned}
$$

Therefore, it suffices to bound the spectral norm $\|\mathbf{E}_{\mathbf{V}\sim U(\mathbf{O}_{n,m})}[\mathbf{V}^{\otimes ak/2}(\mathbf{V}^{\mathsf{T}})^{\otimes ak/2}]\|_2$.

Let $\mathbf{A} = \mathbf{E}_{\mathbf{V}\sim U(\mathbf{O}_{n,m})}[\mathbf{V}^{\otimes ak/2}(\mathbf{V}^{\mathsf{T}})^{\otimes ak/2}]$, $\mathbf{T}_0$ be the eigenvector associated with the largest absolute eigenvalue, and let $\mathbf{u} = \operatorname{argmax}_{\mathbf{u}\in\mathbb{S}^{n-1}}|\langle \mathbf{T}_0, \mathbf{u}^{\otimes ak/2}\rangle|$. Then

$$
\begin{aligned}
\|\mathbf{A}\|_2 &= |\langle \mathbf{A}\mathbf{T}_0, \mathbf{u}^{\otimes ak/2}\rangle|/|\langle \mathbf{T}_0, \mathbf{u}^{\otimes ak/2}\rangle| \\
&= |\langle \mathbf{T}_0, \mathbf{A}\mathbf{u}^{\otimes ak/2}\rangle|/|\langle \mathbf{T}_0, \mathbf{u}^{\otimes ak/2}\rangle| \\
&= |\langle \mathbf{T}_0, \mathbf{E}_{\mathbf{V}\sim U(\mathbf{O}_{n,m})}[(\mathbf{V}\mathbf{V}^{\mathsf{T}}\mathbf{u})^{\otimes ak/2}]\rangle|/|\langle \mathbf{T}_0, \mathbf{u}^{\otimes ak/2}\rangle| \\
&= |\mathbf{E}_{\mathbf{V}\sim U(\mathbf{O}_{n,m})}[\langle \mathbf{T}_0, (\mathbf{V}\mathbf{V}^{\mathsf{T}}\mathbf{u})^{\otimes ak/2}\rangle]|/|\langle \mathbf{T}_0, \mathbf{u}^{\otimes ak/2}\rangle| \\
&\leq \mathbf{E}_{\mathbf{V}\sim U(\mathbf{O}_{n,m})}[|\langle \mathbf{T}_0, (\mathbf{V}\mathbf{V}^{\mathsf{T}}\mathbf{u})^{\otimes ak/2}\rangle|]/|\langle \mathbf{T}_0, \mathbf{u}^{\otimes ak/2}\rangle| \\
&\leq \mathbf{E}_{\mathbf{V}\sim U(\mathbf{O}_{n,m})}[\|(\mathbf{V}\mathbf{V}^{\mathsf{T}}\mathbf{u})^{\otimes ak/2}\|_2 |\langle \mathbf{T}_0, \mathbf{u}^{\otimes ak/2}\rangle|]/|\langle \mathbf{T}_0, \mathbf{u}^{\otimes ak/2}\rangle| \\
&= \mathbf{E}_{\mathbf{V}\sim U(\mathbf{O}_{n,m})}[\|(\mathbf{V}\mathbf{V}^{\mathsf{T}}\mathbf{u})^{\otimes ak/2}\|_2] \\
&= \mathbf{E}_{\mathbf{V}\sim U(\mathbf{O}_{n,m})}\left[\|\mathbf{V}^{\mathsf{T}}\mathbf{u}\|_2^{ak/2}\right] ,
\end{aligned}
$$

where we use $\mathbf{u} = \operatorname{argmax}_{\mathbf{u}\in\mathbb{S}^{n-1}}|\langle \mathbf{T}_0, \mathbf{u}^{\otimes ak/2}\rangle|$ in the second inequality. Using Lemma 3.8, and plug everything back, we get

$$
\mathbf{E}_{\mathbf{V}\sim U(\mathbf{O}_{n,m})}[\|(\mathbf{V}^{\mathsf{T}})^{\otimes k}\mathbf{T}\|_2^a] \leq \mathbf{E}_{\mathbf{V}\sim U(\mathbf{O}_{n,m})}\left[\|\mathbf{V}^{\mathsf{T}}\mathbf{u}\|_2^{ak/2}\right]\|\mathbf{T}\|_2^a .
$$

$\square$

### B.1.6 Proof of Lemma 3.8

*Proof.* To bound $\mathbf{E}_{\mathbf{V}^{\mathsf{T}}\sim U(\mathbf{O}_{n,m})}[\|\mathbf{V}\mathbf{u}\|_2^k]$, by symmetry, we can instead consider $\mathbf{V} = [\mathbf{e}_1, \mathbf{e}_2, \cdots, \mathbf{e}_m]^{\mathsf{T}}$ and $\mathbf{u}\sim U(\mathbb{S}^{n-1})$. Then we have that

$$
\begin{aligned}
\mathbf{E}_{\mathbf{u}\sim U(\mathbb{S}^{n-1})}\left[\|\mathbf{V}\mathbf{u}\|_2^k\right] &= \mathbf{E}_{\mathbf{u}\sim U(\mathbb{S}^{n-1})}\left[(\|\mathbf{V}\mathbf{u}\|_2/\|\mathbf{u}\|_2)^k\right] = \mathbf{E}_{\mathbf{u}\sim U(\mathbb{S}^{n-1})}\left[(\|\mathbf{V}\mathbf{u}\|_2^2/\|\mathbf{u}\|_2^2)^{k/2}\right] \\
&= \mathbf{E}_{t\sim\text{Beta}\left(\frac{m}{2},\frac{n-m}{2}\right)}\left[t^{k/2}\right] ,
\end{aligned}
$$

where the last equation follows from the standard fact that if $X\sim\chi_{d_1}^2$ and $Y\sim\chi_{d_2}^2$ are independent then $\frac{X}{X+Y}\sim\text{Beta}(d_1/2, d_2/2)$. By Stirling's formula, we can bound $\mathbf{E}_{t\sim\text{Beta}\left(\frac{m}{2},\frac{n-m}{2}\right)}\left[t^{k/2}\right]$ as follows:

$$
\begin{aligned}
\mathbf{E}_{t\sim\text{Beta}\left(\frac{m}{2},\frac{n-m}{2}\right)}\left[t^{k/2}\right] &= \frac{1}{\text{B}\left(\frac{m}{2},\frac{n-m}{2}\right)}\int_{t=0}^{1}t^{(\frac{k+m}{2}-1)}(1-t)^{(\frac{n-m}{2}-1)}dt \\
&= \frac{\text{B}\left(\frac{k+m}{2},\frac{n-m}{2}\right)}{\text{B}\left(\frac{m}{2},\frac{n-m}{2}\right)} = \Theta\left(\frac{\Gamma\left(\frac{k+m}{2}\right)\Gamma\left(\frac{n}{2}\right)}{\Gamma\left(\frac{k+n}{2}\right)\Gamma\left(\frac{m}{2}\right)}\right) .
\end{aligned}
$$

$\square$

### B.1.7 Proof of Corollary 3.9

*Proof.* We will discuss by cases. We first consider the case where both $n$ and $m$ are even integers. For $k \leq n^c$, by definition of $\Gamma$ function, we have that

$$
\begin{aligned}
\mathbf{E}_{\mathbf{V} \sim U(\mathbf{O}_{n,m})}[\|\mathbf{V}^\intercal \mathbf{u}\|_2^k] &= \Theta \left( \frac{\Gamma\left(\frac{k+m}{2}\right) \Gamma\left(\frac{n}{2}\right)}{\Gamma\left(\frac{k+n}{2}\right) \Gamma\left(\frac{m}{2}\right)} \right) \\
&= \Theta \left( \frac{\left(\frac{n}{2} - 1\right)\left(\frac{n}{2} - 2\right) \cdots \left(\frac{m}{2}\right)}{\left(\frac{k+n}{2} - 1\right)\left(\frac{k+n}{2} - 2\right) \cdots \left(\frac{k+m}{2}\right)} \right) \\
&= \Theta \left( \frac{\left(\frac{k+m}{2} - 1\right)\left(\frac{k+m}{2} - 2\right) \cdots \left(\frac{m}{2}\right)}{\left(\frac{k+n}{2} - 1\right)\left(\frac{k+n}{2} - 2\right) \cdots \left(\frac{n}{2}\right)} \right) \\
&\leq O \left( \left(\frac{k+m}{k+n}\right)^{k/2} \right) \\
&= O(2^{k/2} n^{-(1-c)k/2}) .
\end{aligned}
\tag{2}
$$

For $k \geq n^c$, we have that

$$
\begin{aligned}
\mathbf{E}_{\mathbf{V} \sim U(\mathbf{O}_{n,m})}[\|\mathbf{V}^\intercal \mathbf{u}\|_2^k] &= \Theta \left( \frac{\Gamma\left(\frac{k+m}{2}\right) \Gamma\left(\frac{n}{2}\right)}{\Gamma\left(\frac{k+n}{2}\right) \Gamma\left(\frac{m}{2}\right)} \right) \\
&= \Theta \left( \frac{\left(\frac{n}{2} - 1\right)\left(\frac{n}{2} - 2\right) \cdots \left(\frac{m}{2}\right)}{\left(\frac{n^c+n}{2} - 1\right)\left(\frac{n^c+n}{2} - 2\right) \cdots \left(\frac{n^c+m}{2}\right)} \times \frac{\left(\frac{n^c+n}{2} - 1\right)\left(\frac{n^c+n}{2} - 2\right) \cdots \left(\frac{n^c+m}{2}\right)}{\left(\frac{k+n}{2} - 1\right)\left(\frac{k+n}{2} - 2\right) \cdots \left(\frac{k+m}{2}\right)} \right) \\
&= O(2^{n^c/2} n^{-(1-c)n^c/2}) O \left( \left(\frac{n^c+n}{k+n}\right)^{(n-m)/2} \right) \\
&= \exp(-\Omega(n^c)) O \left( \left(\frac{n^c+n}{k+n}\right)^{(n-m)/2} \right) ,
\end{aligned}
\tag{3}
$$

where the third equation follows from equation (2) by taking $k = n^c$.

For the case where $n$ is even and $m$ is odd, note that $\Gamma(x + 1/2) = \Theta(\sqrt{x}\Gamma(x))$ by Stirling approximation, we have that

$$
\mathbf{E}_{\mathbf{V} \sim U(\mathbf{O}_{n,m})}[\|\mathbf{V}^\intercal \mathbf{u}\|_2^k] = \Theta \left( \frac{\Gamma\left(\frac{k+m}{2}\right) \Gamma\left(\frac{n}{2}\right)}{\Gamma\left(\frac{k+n}{2}\right) \Gamma\left(\frac{m}{2}\right)} \right) = \Theta \left( \frac{\sqrt{m}\Gamma\left(\frac{k+m-1}{2}\right) \Gamma\left(\frac{n}{2}\right)}{\sqrt{k+m}\Gamma\left(\frac{k+n}{2}\right) \Gamma\left(\frac{m-1}{2}\right))} \right) .
$$

For $k \leq n^c$, by equation 2, we have that

$$
\mathbf{E}_{\mathbf{V} \sim U(\mathbf{O}_{n,m})}[\|\mathbf{V}^\intercal \mathbf{u}\|_2^k] = \Theta \left( \frac{\sqrt{m}\Gamma\left(\frac{k+m+1}{2}\right) \Gamma\left(\frac{n}{2}\right)}{\sqrt{k+m}\Gamma\left(\frac{k+n}{2}\right) \Gamma\left(\frac{m+1}{2}\right))} \right) = O \left( 2^{k/2} n^{-(1-c)k/2} \right) .
$$

For $k \geq n^c$, by equation 3, we have that

$$
\begin{aligned}
\mathbf{E}_{\mathbf{V} \sim U(\mathbf{O}_{n,m})}[\|\mathbf{V}^\intercal \mathbf{u}\|_2^k] &= \Theta \left( \frac{\sqrt{m}\Gamma\left(\frac{k+m-1}{2}\right) \Gamma\left(\frac{n}{2}\right)}{\sqrt{k+m}\Gamma\left(\frac{k+n}{2}\right) \Gamma\left(\frac{m-1}{2}\right))} \right) \\
&= \exp(-\Omega(n^c)) O \left( \left(\frac{n^c+n}{k+n}\right)^{(n-m+1)/2} \right) \\
&\leq \exp(-\Omega(n^c)) O \left( \left(\frac{n^c+n}{k+n}\right)^{(n-m)/2} \right) .
\end{aligned}
$$

For the case where $n$ is odd and $m$ is even, by Stirling approximation, we have that

$$
\mathbf{E}_{\mathbf{V} \sim U(\mathbf{O}_{n,m})}[\|\mathbf{V}^\intercal \mathbf{u}\|_2^k] = \Theta \left( \frac{\Gamma\left(\frac{k+m}{2}\right) \Gamma\left(\frac{n}{2}\right)}{\Gamma\left(\frac{k+n}{2}\right) \Gamma\left(\frac{m}{2}\right)} \right) = \Theta \left( \frac{\sqrt{k+n}\Gamma\left(\frac{k+m}{2}\right) \Gamma\left(\frac{n+1}{2}\right)}{\sqrt{n}\Gamma\left(\frac{k+n+1}{2}\right) \Gamma\left(\frac{m}{2}\right))} \right) .
$$

For $k \leq n^c$, by equation 2, we have that

$$\mathbf{E}_{\mathbf{V} \sim U(\mathbf{O}_{n,m})}[\|\mathbf{V}^{\mathsf{T}}\mathbf{u}\|_2^k] = \Theta\left(\frac{\sqrt{k+n}\,\Gamma\left(\frac{k+m}{2}\right)\Gamma\left(\frac{n+1}{2}\right)}{\sqrt{n}\,\Gamma\left(\frac{k+n+1}{2}\right)\Gamma\left(\frac{m}{2}\right))}\right)$$

$$= O\left(\sqrt{\frac{k+n}{n}}2^{k/2}(n+1)^{-(1-c)k/2}\right)$$

$$= O\left(2^{k/2}n^{-(1-c)k/2}\right).$$

For $k \geq n^c$, by equation 3, we have that

$$\mathbf{E}_{\mathbf{V} \sim U(\mathbf{O}_{n,m})}[\|\mathbf{V}^{\mathsf{T}}\mathbf{u}\|_2^k] = \Theta\left(\frac{\sqrt{k+n}\,\Gamma\left(\frac{k+m}{2}\right)\Gamma\left(\frac{n+1}{2}\right)}{\sqrt{n}\,\Gamma\left(\frac{k+n+1}{2}\right)\Gamma\left(\frac{m}{2}\right))}\right)$$

$$= \exp(-\Omega(n^c))O\left(\sqrt{\frac{k+n}{n}}\left(\frac{n^c+n+1}{k+n+1}\right)^{(n-m+1)/2}\right)$$

$$\leq \exp(-\Omega(n^c))O\left(\left(\frac{n^c+n}{k+n}\right)^{(n-m)/2}\right).$$

For the case where both $n$ and $m$ are odd integers, by Stirling approximation, we have that

$$\mathbf{E}_{\mathbf{V} \sim U(\mathbf{O}_{n,m})}[\|\mathbf{V}^{\mathsf{T}}\mathbf{u}\|_2^k] = \Theta\left(\frac{\Gamma\left(\frac{k+m}{2}\right)\Gamma\left(\frac{n}{2}\right)}{\Gamma\left(\frac{k+n}{2}\right)\Gamma\left(\frac{m}{2}\right)}\right) = \Theta\left(\frac{\sqrt{m(k+n)}\,\Gamma\left(\frac{k+m-1}{2}\right)\Gamma\left(\frac{n+1}{2}\right)}{\sqrt{n(k+m)}\,\Gamma\left(\frac{k+n+1}{2}\right)\Gamma\left(\frac{m-1}{2}\right))}\right).$$

For $k \leq n^c$, by equation 2, we have that

$$\mathbf{E}_{\mathbf{V} \sim U(\mathbf{O}_{n,m})}[\|\mathbf{V}^{\mathsf{T}}\mathbf{u}\|_2^k] = \Theta\left(\frac{\sqrt{m(k+n)}\,\Gamma\left(\frac{k+m-1}{2}\right)\Gamma\left(\frac{n+1}{2}\right)}{\sqrt{n(k+m)}\,\Gamma\left(\frac{k+n+1}{2}\right)\Gamma\left(\frac{m-1}{2}\right))}\right)$$

$$= O\left(\sqrt{\frac{m(k+n)}{n(k+m)}}2^{k/2}(n+1)^{-(1-c)k/2}\right)$$

$$= O\left(2^{k/2}n^{-(1-c)k/2}\right).$$

For $k \geq n^c$, by equation 3, we have that

$$\mathbf{E}_{\mathbf{V} \sim U(\mathbf{O}_{n,m})}[\|\mathbf{V}^{\mathsf{T}}\mathbf{u}\|_2^k] = \Theta\left(\frac{\sqrt{m(k+n)}\,\Gamma\left(\frac{k+m-1}{2}\right)\Gamma\left(\frac{n+1}{2}\right)}{\sqrt{n(k+m)}\,\Gamma\left(\frac{k+n+1}{2}\right)\Gamma\left(\frac{m-1}{2}\right))}\right)$$

$$= \exp(-\Omega(n^c))O\left(\sqrt{\frac{m(k+n)}{n(k+m)}}\left(\frac{n^c+n+1}{k+n+1}\right)^{(n-m+2)/2}\right)$$

$$\leq \exp(-\Omega(n^c))O\left(\left(\frac{n^c+n}{k+n}\right)^{(n-m)/2}\right).$$

$\square$

### B.1.8   Proof of Lemma 3.10

*Proof.* As we have discussed, we will consider three ranges of $k$. However, for some technical reasons and the ease of calculations, we will additionally break the second range into two ranges. We can write

$$\sum_{k=1}^{\ell}|\langle\mathbf{A}_k,(\mathbf{V}^{\mathsf{T}})^{\otimes k}\mathbf{T}_k\rangle| = \sum_{k=1}^{d-1}|\langle\mathbf{A}_k,(\mathbf{V}^{\mathsf{T}})^{\otimes k}\mathbf{T}_k\rangle| + \sum_{k=d}^{n^\lambda}|\langle\mathbf{A}_k,(\mathbf{V}^{\mathsf{T}})^{\otimes k}\mathbf{T}_k\rangle|$$

$$+ \sum_{k=n^\lambda+1}^{T}|\langle\mathbf{A}_k,(\mathbf{V}^{\mathsf{T}})^{\otimes k}\mathbf{T}_k\rangle| + \sum_{k=T+1}^{\ell}|\langle\mathbf{A}_k,(\mathbf{V}^{\mathsf{T}})^{\otimes k}\mathbf{T}_k\rangle|,$$

where $T$ is a value we will later specify. To analyze each $|\langle \mathbf{A}_k, (\mathbf{V}^\intercal)^{\otimes k} \mathbf{T}_k \rangle|$, recall that $|\langle \mathbf{A}_k, (\mathbf{V}^\intercal)^{\otimes k} \mathbf{T}_k \rangle| \leq \|\mathbf{A}_k\|_2 \|(\mathbf{V}^\intercal)^{\otimes k} \mathbf{T}_k\|_2$, where $\mathbf{A}_k = \mathbf{E}_{\mathbf{x} \sim A'}[\mathbf{H}_k(\mathbf{x})]$ is a constant (not depending on the randomness of $\mathbf{V}$). For $\|(\mathbf{V}^\intercal)^{\otimes k} \mathbf{T}_k\|_2$, we can show it is small by bounding its $a$-th moment for even $a$ using Lemma 3.7 which says

$$\mathbf{E}_{\mathbf{V} \sim U(\mathbf{O}_{n,m})}[\|(\mathbf{V}^\intercal)^{\otimes k} \mathbf{T}\|_2^a] \leq \mathbf{E}_{\mathbf{V} \sim U(\mathbf{O}_{n,m})} \left[\|\mathbf{V}^\intercal \mathbf{u}\|_2^{ak/2}\right] \|\mathbf{T}\|_2^a \,,$$

for some unit vector $\mathbf{u} \in \mathbb{S}^{n-1}$. We will apply this strategy on the four different ranges of $k$. We start by picking the following parameters (the sufficiently close here only depends on $c$):

- We require $d, m \leq n^\lambda / \log n$;

- $B = n^\alpha$ where $\alpha < (1 - \lambda)/8$ is sufficiently close;

- $T = n^\beta$ where $\beta > (1 - \lambda)/4$ is sufficiently close;

- We let $\lambda_3 > \lambda_2 > \lambda_1 > \lambda$ to be sufficiently close (the difference between these quantities will be a sufficient small constant fraction of $c$);

- We let $\beta_3 > \beta_2 > \beta_1 > \beta$ to be sufficiently close (the difference between these quantities will be a sufficient small constant fraction of $c$).

WLOG, we will assume $\lambda \geq 8c$. Suppose $\lambda \geq 8c$ is not true, then we can simply consider a new pair $\lambda', c'$, where $\lambda' = \lambda + 4c$ and $c' = c/2$. Notice that $(1 - \lambda)/8 - c = (1 - \lambda')/8 - c'$, therefore, the SQ lower bound in the statement remains unchange. For convenience, we let $\zeta = (1 - \lambda)/8 - c$. WLOG, we can assume $1 < d$, $d < n^\lambda$, $n^\lambda < T$ and $T < \ell$ for each case, because otherwise, our upper bounds still bounds the value of $\sum_{k=1}^{\ell} |\langle \mathbf{A}_k, (\mathbf{V}^\intercal)^{\otimes k} \mathbf{T}_k \rangle|$. We also always assume $n$ is at least a sufficiently large integer depending on $(1 - \lambda)/8 - \zeta$. In order to apply Lemma 3.2, we need to first check that $B$ satisfies the condition that $B^d \geq c_1 \left(2^{d/2} \sqrt{\frac{\Gamma(d+m/2)}{\Gamma(m/2)}}\right)$, where $c_1$ is at least a sufficiently large universal constant.

We first note that since

(i) $\left(\frac{\Gamma(d/2+m/2)}{\Gamma(m/2)}\right) n^{-\zeta d} < 2$,

(ii) $\zeta \leq (1 - \lambda)/8$,

(iii) $\frac{\Gamma(d/2+m/2)}{\Gamma(m/2)} \geq 2^{-O(d)} \sqrt{\frac{\Gamma(d+m/2)}{\Gamma(m/2)}}$,

(iv) $n$ is at least a sufficiently large integer depending on $(1 - \lambda)/8 - \zeta$, and

(v) $\alpha < (1 - \lambda)/8$ is sufficiently close depending on $(1 - \lambda)/8 - \zeta$,

we have $\sqrt{\frac{\Gamma(d+m/2)}{\Gamma(m/2)}} B^{-d} \leq 2^{O(d)} n^{-(\alpha-\zeta)} \leq \log^{-1} n$. Given $n$ is sufficiently large, $B$ satisfies the condition in Lemma 3.2. This implies the following bound on total variation distance:

$$d_{\mathrm{TV}}(A, A') \leq O \left(2^{d/2} \sqrt{\frac{\Gamma(d+m/2)}{\Gamma(m/2)}}\right) B^{-d} \leq \left(\frac{\Gamma(d/2+m/2)}{\Gamma(m/2)}\right) n^{-\zeta d} \,.$$

We now bound the summation $\sum_{k=1}^{\ell} |\langle \mathbf{A}_k, (\mathbf{V}^\intercal)^{\otimes k} \mathbf{T}_k \rangle|$ as follows:

- $\sum_{k=1}^{d-1} |\langle \mathbf{A}_k, (\mathbf{V}^\intercal)^{\otimes k} \mathbf{T}_k \rangle|$ **is small with high probability:** Since $\left(\sqrt{\frac{\Gamma(d+m/2)}{\Gamma(m/2)}}\right) n^{-\zeta d} < 2$, $\zeta \leq (1 - \lambda)/8$ and $B = n^\alpha$, where $\alpha$ is sufficiently close to $(1 - \lambda)/8$ and the parameters satisfies the condition $B^d \geq 6 \left(2^{d/2} \sqrt{\frac{\Gamma(d+m/2)}{\Gamma(m/2)}}\right)$ in Lemma 3.2. Since $k < d$, by Lemma 3.2, we

have $\|\mathbf{A}_k\|_2 = 2^{O(k)} \left(2^{d/2}\sqrt{\frac{\Gamma(d+m/2)}{\Gamma(m/2)}}\right) B^{-(d-k)} + \nu$. Let $a$ be the largest even number that $ak/2 \leq n^\lambda$, where $d = o(n^\lambda)$ implies $a \geq 2$. Then using Lemma 3.7 and Lemma 3.8, we have

$$
\begin{aligned}
\mathbf{E}_{\mathbf{V}\sim U(\mathbf{O}_{n,m})}[\|(\mathbf{V}^\intercal)^{\otimes k}\mathbf{T}_k\|_2^a] &= \mathbf{E}_{\mathbf{V}\sim U(\mathbf{O}_{n,m})}\left[\|\mathbf{V}^\intercal \mathbf{u}\|_2^{ak/2}\right]\|\mathbf{T}_k\|_2^a \\
&\leq \mathbf{E}_{\mathbf{V}\sim U(\mathbf{O}_{n,m})}\left[\|\mathbf{V}^\intercal \mathbf{u}\|_2^{ak/2}\right] \\
&= O(2^{ak/4} n^{-(1-\lambda)ak/4}) \\
&= O(n^{-(1-\lambda_1)ak/4}) \ .
\end{aligned}
$$

Using Markov's Inequality, this implies the tail bound

$$
\mathbf{Pr}[\|(\mathbf{V}^\intercal)^{\otimes k}\mathbf{T}_k\|_2 \geq n^{-(1-\lambda_2)k/4}] \leq 2^{-\Omega(cn^\lambda)} = 2^{-n^{\Omega(c)}} \ .
$$

Therefore, we have

$$
\begin{aligned}
\sum_{k=1}^{d-1} |\langle \mathbf{A}_k, (\mathbf{V}^\intercal)^{\otimes k}\mathbf{T}_k\rangle| &\leq \sum_{k=1}^{d-1} \|\mathbf{A}_k\|_2 \|(\mathbf{V}^\intercal)^{\otimes k}\mathbf{T}_k\|_2 \\
&\leq \sum_{k=1}^{d-1} n^{-(1-\lambda_2)k/4}\left(2^{O(k)}\left(2^{d/2}\sqrt{\frac{\Gamma(d+m/2)}{\Gamma(m/2)}}\right) B^{-(d-k)} + \nu\right) \\
&\leq (1+o(1))\left(2^{d/2}\sqrt{\frac{\Gamma(d+m/2)}{\Gamma(m/2)}}B^{-d} + \nu\right) \\
&= (1+o(1))\left(2^{d/2}\sqrt{\frac{\Gamma(d+m/2)}{\Gamma(m/2)}}n^{-\alpha d} + \nu\right) \ ,
\end{aligned}
$$

except with probability $2^{-n^{\Omega(c)}}$, where the equation above follows from $B = n^\alpha = o(n^{(1-\lambda)/8}) = o(n^{(1-\lambda_2)/4})$.

- $\sum_{k=d}^{n^\lambda} |\langle \mathbf{A}_k, (\mathbf{V}^\intercal)^{\otimes k}\mathbf{T}_k\rangle|$ **is small with high probability:** In the previous case, we have argued $B^d \geq 6\left(2^{d/2}\sqrt{\frac{\Gamma(d+m/2)}{\Gamma(m/2)}}\right)$. This implies that $B^2 \geq m$. Therefore, combining Fact 3.5 with the fact that $A'$ is bounded inside $\mathbb{B}^m(B)$, we have that

$$
\|\mathbf{A}_k\|_2 = \|\mathbf{E}_{\mathbf{x}\sim A'}[\mathbf{H}_k(\mathbf{x})] - \mathbf{E}_{\mathbf{x}\sim\mathcal{N}_m}[\mathbf{H}_k(\mathbf{x})]\|_2 = \|\mathbf{E}_{\mathbf{x}\sim A'}[\mathbf{H}_k(\mathbf{x})]\|_2 \leq 2^{O(k)}B^k \ .
$$

Let $a$ be the largest even number that $ak/2 \leq n^\lambda$, where $d = o(n^\lambda)$ implies $a \geq 2$. The using the same argument gives

$$
\mathbf{E}_{\mathbf{V}\sim U(\mathbf{O}_{n,m})}[\|(\mathbf{V}^\intercal)^{\otimes k}\mathbf{T}_k\|_2^a] = O(n^{-(1-\lambda_1)ak/4}) \ .
$$

Using Markov's Inequality, this implies the tail bound

$$
\mathbf{Pr}[\|(\mathbf{V}^\intercal)^{\otimes k}\mathbf{T}_k\|_2 \geq n^{-(1-\lambda_2)k/4}] \leq 2^{-\Omega(cn^\lambda)} = 2^{-n^{\Omega(c)}} \ .
$$

Therefore, we have

$$
\begin{aligned}
\sum_{k=d}^{n^\lambda} |\langle \mathbf{A}_k, (\mathbf{V}^\intercal)^{\otimes k}\mathbf{T}_k\rangle| &\leq \sum_{k=d}^{n^\lambda} \|\mathbf{A}_k\|_2 \|(\mathbf{V}^\intercal)^{\otimes k}\mathbf{T}_k\|_2 \\
&\leq \sum_{k=d}^{n^\lambda} n^{-(1-\lambda_2)k/4} 2^{O(k)} B^k \\
&= 2^{O(d)} n^{-((1-\lambda_2)/4-\alpha)d} \\
&= n^{-((1-\lambda_3)/4-\alpha)d} \ ,
\end{aligned}
$$

except with probability $2^{-n^{\Omega(c)}}$ (the first equality above follows from $B = n^\alpha = o(n^{(1-\lambda)/8}) = o(n^{(1-\lambda_2)/4})$).

- $\sum_{k=n^\lambda+1}^{T}|\langle\mathbf{A}_k,(\mathbf{V}^\intercal)^{\otimes k}\mathbf{T}_k\rangle|$ **is small with high probability:** We can WLOG assume $n^\lambda < T$, because otherwise, this term is 0. Same as the above case, we have

$$\|\mathbf{A}_k\|_2 = \|\mathbf{E}_{\mathbf{x}\sim A'}[\mathbf{H}_k(\mathbf{x})] - \mathbf{E}_{\mathbf{x}\sim\mathcal{N}_m}[\mathbf{H}_k(\mathbf{x})]\|_2 = \|\mathbf{E}_{\mathbf{x}\sim A'}[\mathbf{H}_k(\mathbf{x})]\|_2 \leq 2^{O(k)}B^k .$$

Then let $a$ be the largest even number that $ak/2 \leq T$, where $k \leq T$ implies $a \geq 2$. Since $ak/2 \leq n^\beta$ and $m \leq n^\lambda < T = n^\beta$, the same argument implies

$$\mathbf{E}_{\mathbf{V}\sim U(\mathbf{O}_{n,m})}[\|(\mathbf{V}^\intercal)^{\otimes k}\mathbf{T}_k\|_2^a] = O(n^{-(1-\beta_1)ak/4}) ,$$

and implies the tail bound

$$\mathbf{Pr}[\|(\mathbf{V}^\intercal)^{\otimes k}\mathbf{T}_k\|_2 \geq n^{-(1-\beta_2)k/4}] \leq 2^{-\Omega(cT)} = 2^{-n^{\Omega(c)}} .$$

Given $n^\lambda < T = n^\beta$ and $\beta > (1-\lambda)/4$ is sufficiently close, we have $(1-\beta_3)/4$ is at least $1/6$, therefore,

$$\begin{aligned}
\sum_{k=n^\lambda+1}^{T}|\langle\mathbf{A}_k,(\mathbf{V}^\intercal)^{\otimes k}\mathbf{T}_k\rangle| &\leq \sum_{k=n^\lambda+1}^{T}\|\mathbf{A}_k\|_2\|(\mathbf{V}^\intercal)^{\otimes k}\mathbf{T}_k\|_2\\
&\leq \sum_{k=n^\lambda+1}^{T}n^{-(1-\beta_2)k/4}2^{O(k)}B^k\\
&= n^{-(1-\beta_2)n^\lambda/4}2^{O(n^\lambda)}B^{n^\lambda}\\
&= n^{-(1-\beta_3)n^\lambda/4}B^{n^\lambda}\\
&= n^{-((1-\beta_3)/4-\alpha)d\log n}\\
&= n^{-d} ,
\end{aligned}$$

except with probability $2^{-n^{\Omega(c)}}$, where the first equation above follows from $B = n^\alpha = o(n^{1/8}) = o(n^{(1-\beta_2)/4})$, the third equation follows from $n^\lambda \geq d\log n$ and the last equation follows from the assumption that $n$ is at least a sufficiently large integer and $(1-\beta_3)/4 - \alpha \geq 1/6 - 1/8)$.

- $\sum_{k=T+1}^{\ell}|\langle\mathbf{A}_k,(\mathbf{V}^\intercal)^{\otimes k}\mathbf{T}_k\rangle|$ **is small with high probability:** Combining Fact 3.6 with the fact that $A'$ is bounded inside $\mathbb{B}^m(B)$, we have that

$$\begin{aligned}
\|\mathbf{A}_k\|_2 &= \|\mathbf{E}_{\mathbf{x}\sim A'}[\mathbf{H}_k(\mathbf{x})] - \mathbf{E}_{\mathbf{x}\sim\mathcal{N}_m}[\mathbf{H}_k(\mathbf{x})]\|_2 = \|\mathbf{E}_{\mathbf{x}\sim A'}[\mathbf{H}_k(\mathbf{x})]\|_2\\
&\leq 2^{O(m)}\binom{k+m-1}{m-1}^{1/2}\exp(B^2/4) .
\end{aligned}$$

We take $a = 2$. Note that $ak/2 > T = n^\beta$, applying Lemma 3.7 and Lemma 3.8 yields

$$\begin{aligned}
\mathbf{E}_{\mathbf{V}\sim U(\mathbf{O}_{n,m})}[\|(\mathbf{V}^\intercal)^{\otimes k}\mathbf{T}_k\|_2^a] &= \mathbf{E}_{\mathbf{V}\sim U(\mathbf{O}_{n,m})}\left[\|\mathbf{V}^\intercal\mathbf{u}\|_2^{ak/2}\right]\|\mathbf{T}_k\|_2^a\\
&= \mathbf{E}_{\mathbf{V}\sim U(\mathbf{O}_{n,m})}\left[\|\mathbf{V}^\intercal\mathbf{u}\|_2^{ak/2}\right]\\
&= \exp(-\Omega(n^\beta))O\left(\left(\frac{n^\beta+n}{k+n}\right)^{(n-m)/2}\right) .
\end{aligned}$$

Applying Markov's inequality yields the tail bound

$$\mathbf{Pr}\left[\|(\mathbf{V}^\intercal)^{\otimes k}\mathbf{T}_k\|_2 \geq 2^{-\Omega(n^\beta)}O\left(\left(\frac{n^\beta+n}{k+n}\right)^{(n-m)/4}\right)\right] \leq 2^{-\Omega(n^\beta)} = 2^{-\Omega(n^{(1-\lambda)/4})} = 2^{-n^{\Omega(c)}} .$$

Therefore, we have

$$\sum_{k=T+1}^{\ell} |\langle \mathbf{A}_k, (\mathbf{V}^\intercal)^{\otimes k} \mathbf{T}_k \rangle|$$

$$\leq \sum_{k=T+1}^{\infty} |\langle \mathbf{A}_k, (\mathbf{V}^\intercal)^{\otimes k} \mathbf{T}_k \rangle| \leq \sum_{k=T+1}^{\infty} \|\mathbf{A}_k\|_2 \|(\mathbf{V}^\intercal)^{\otimes k} \mathbf{T}_k\|_2$$

$$\leq \sum_{k=T+1}^{\infty} 2^{O(m)} \binom{k+m-1}{m-1}^{1/2} \exp(B^2/4) 2^{-\Omega(n^\beta)} O\left( \left( \frac{n^\beta + n}{k+n} \right)^{(n-m)/4} \right)$$

$$\leq \sum_{k=T}^{\infty} 2^{-\Omega(n^\beta)} \left( \frac{k+m}{T+m} \right)^{m/2} \left( \frac{T+n}{k+n} \right)^{n/8} ,$$

where the last inequality follows from our choice of parameters. Therefore, we have that

$$\sum_{k=T+1}^{\ell} |\langle \mathbf{A}_k, (\mathbf{V}^\intercal)^{\otimes k} \mathbf{T}_k \rangle| \leq \sum_{k=T}^{\infty} 2^{-\Omega(n^\beta)} \left( 1 + \frac{k-T}{T+m} \right)^{m/2} \left( 1 + \frac{k-T}{T+n} \right)^{-n/8}$$

$$\leq \sum_{k=T}^{\infty} 2^{-\Omega(n^\beta)} \left( 1 + \frac{k-T}{T+n} \right)^{(m/2)(2n/T)} \left( 1 + \frac{k-T}{T+n} \right)^{-n/8}$$

$$\leq \sum_{k=T}^{\infty} 2^{-\Omega(n^\beta)} \left( 1 + \frac{k-T}{T+n} \right)^{-n/8 + n^{1+\gamma-\beta}}$$

$$\leq \sum_{k=T}^{\infty} 2^{-\Omega(n^\beta)} \left( \frac{T+n}{k+n} \right)^{n/16}$$

$$\leq 2^{-\Omega(n^\beta)} \int_{k=T-1}^{\infty} \left( \frac{T+n}{k+n} \right)^{n/16} dk$$

$$= \frac{2^{-\Omega(n^\beta)}(T+n-1)(1+1/(T+n-1))^{n/16}}{n/16-1}$$

$$= 2^{-\Omega(n^\beta)} ,$$

except with probability $2^{-n^{\Omega(c)}}$.

Adding the three cases above together, we get for any $d, m \leq n^\lambda / \log n$, $\zeta = (1-\lambda)/8 - c$ where $c > 0$ and $n$ is at least a sufficiently small constant depending on $c$,

$$\sum_{k=1}^{\ell} |\langle \mathbf{A}_k, (\mathbf{V}^\intercal)^{\otimes k} \mathbf{T}_k \rangle|$$

$$\leq (1+o(1)) \left( 2^{d/2} \sqrt{\frac{\Gamma(d+m/2)}{\Gamma(m/2)}} n^{-\alpha d} + \nu \right) + n^{-((1-\lambda_3)/4-\alpha)d} + n^{-d} + 2^{-\Omega(n^\beta)}$$

$$\leq \left( \frac{\Gamma(d/2+m/2)}{\Gamma(m/2)} \right) n^{-\zeta d} + (1+o(1))\nu$$

$$= \left( \frac{\Gamma(d/2+m/2)}{\Gamma(m/2)} \right) n^{-((1-\lambda)/8-c)d} + (1+o(1))\nu ,$$

except with probability $2^{-n^{\Omega(c)}}$, where the second from the last inequality above follows from $\frac{\Gamma(d/2+m/2)}{\Gamma(m/2)} \geq 2^{-O(d)} \sqrt{\frac{\Gamma(d+m/2)}{\Gamma(m/2)}}$. $\qquad\square$

## B.2 Omitted Proofs from Section 3.2

### B.2.1 Proof of Lemma 3.11

*Proof.* Notice that the distribution $D$ is a symmetric distribution. Thus, if we can show that for $\mathbf{x} \sim D$ the distribution $\|\mathbf{x}\|_2^2$ is a continuous distribution, then $D$ is also a continuous distribution. Note that the distribution $D$ can be thought of as generated by the following process. To generate $\mathbf{x} \sim D$, we first sample $\mathbf{t} \sim A'$, $\mathbf{x}' \sim \mathcal{N}_{n-m}$. Let $\mathbf{u}_1, \cdots, \mathbf{u}_{n-m}$ be an orthonormal basis that spans the orthogonal complement of $\text{span}(\mathbf{V})$. We let $\mathbf{x} = \sum_{i=1}^m t_i \mathbf{v}_i + \sum_{i=1}^{d-1} \mathbf{x}_i \mathbf{u}_i$. Noting that $\|\mathbf{x}\|_2^2 = \|\mathbf{t}\|_2^2 + \|\mathbf{x}'\|_2^2$, its distribution is the convolution sum of the distribution of $\|\mathbf{t}\|_2^2$ and the $\chi_{n-m}^2$ distribution, which is continuous. Thus, as argued above, $D$ is a continuous distribution.

It remains to argue that $\chi^2(D, \mathcal{N}_n) = O_n(1)$. We use $D_{r^2}$ to denote the distribution of $\|\mathbf{x}\|_2^2$ above and $P_{r^2}$ to denote its pdf. We use $S_{n-1}(r\mathbb{S}^{n-1})$ to denote the surface area of the $n$-dimensional sphere $r\mathbb{S}^{n-1}$ with radius $r$. Then the pdf function of $D$ is

$$D(\mathbf{x}) = P_{r^2}(\|\mathbf{x}\|_2^2)/S_{n-1}(\|\mathbf{x}\|_2 \mathbb{S}^{n-1}) \ .$$

Similarly, the pdf function of $\mathcal{N}_n$ is

$$\mathcal{N}_n(\mathbf{x}) = \chi_n^2(\|\mathbf{x}\|_2^2)/S_{n-1}(\|\mathbf{x}\|_2 \mathbb{S}^{n-1}) \ .$$

Then we have

$$
\begin{aligned}
1 + \chi^2(D, \mathcal{N}_n) &= \int_{\mathbb{R}^n} \frac{D(\mathbf{x})^2}{\mathcal{N}_n(\mathbf{x})} d\mathbf{x} \\
&= \int_{\mathbb{R}^n} \frac{P_{r^2}(\|\mathbf{x}\|_2^2)^2}{\chi_n^2(\|\mathbf{x}\|_2^2) S_{n-1}(\|\mathbf{x}\|_2 \mathbb{S}^{n-1})} d\mathbf{x} \\
&= \int_0^\infty \int_{r\mathbb{S}^{n-1}} \frac{P_{r^2}(\|\mathbf{x}\|_2^2)^2}{\chi_n^2(\|\mathbf{x}\|_2^2) S_{n-1}(\|\mathbf{x}\|_2 \mathbb{S}^{n-1})} d\mathbf{x} dr \\
&= \int_0^\infty \frac{P_{r^2}(r^2)^2}{\chi_n^2(r^2)} dr \\
&= \int_0^\infty \frac{P_{r^2}(r)^2}{2\sqrt{r} \chi_n^2(r)} dr \ .
\end{aligned}
$$

Thus, it remains to show that $\int_0^\infty \frac{P_{r^2}(r)^2}{2\sqrt{r}\chi_n^2(r)} dr = O_n(1)$.

We will first give a pointwise upper bound on $P_{r^2}$. Notice that $D_{r^2}$ is the convolution sum of $D_{\|\mathbf{t}\|_2^2}$ and the $\chi_{n-m}^2$ distribution, where $\|\mathbf{t}\|_2^2$ is inside $[0, n^2]$ since $A$ is supported on $\mathbb{B}^{m-1}(n)$. Thus, we can write

$$P_{r^2}(r) = \int_0^{n^2} P_{\|\mathbf{t}\|_2^2}(s) \chi_{n-m}^2(r-s) ds \leq \max_{s \in [r-n^2, r]} \chi_{n-m}^2(s) \ .$$

Plugging the pointwise upper bound back, we get

$$
\begin{aligned}
1 + \chi^2(D, \mathcal{N}_n) &\leq \int_0^\infty \frac{\left(\max_{s \in [r-n^2, r]} \chi_{n-m}^2(s)\right)^2}{2\sqrt{r}\chi_n^2(r)} dr \\
&= O_n(1) \int_0^\infty \frac{\left(\max_{s \in [r-n^2, r]} s^{(n-m)/2-1} e^{-s/2}\right)^2}{r^{n/2-1/2} e^{-r/2}} dr \\
&= O_n(1) \int_0^\infty \frac{\max_{s \in [r-n^2, r]} s^{n-m-2} e^{-s}}{r^{n/2-1/2} e^{-r/2}} dr \\
&\leq O_n(1) \int_0^\infty r^{n/2-m-3/2} e^{-r/2} e^{n^2} dr \\
&= O_n(1) \int_0^\infty r^{n/2-m-3/2} e^{-r/2} dr \\
&= O_n(1) \Gamma(n/2 - m - 1/2) = O_n(1) \ .
\end{aligned}
$$

This completes the proof. $\qquad\square$

# C Omitted Details on Applications

In this section, we provide additional context on our applications and provide the proofs of Theorems 1.8 and 1.10.

## C.1 Proof of Theorem 1.8

*Proof.* This is a direct application of Theorem 1.5. We will let the one-dimensional moment-matching distribution be the distribution in Fact 1.7. Then any SQ algorithm distinguishing between

- A standard Gaussian; and
- The distribution $\mathbf{P}_{\mathbf{v}}^A$ for $\mathbf{v} \sim U(\mathbb{S}^{n-1})$, where $A = \alpha \mathcal{N}(\mu, 1) + (1-\alpha)E$ and $\mu = 10 c_d \alpha^{-1/d}$,

with at least $2/3$ probability must require either a query of tolerance at most $O_d(n^{-((1-\lambda)/8-c)d})$ or $2^{n^{\Omega(1)}}$ many queries. This proves the SQ lower bound for the NGCA testing problem. However, it is not clear if there is a simple and optimal reduction from list-decodable Gaussian estimation to the hypothesis testing problem. Therefore, we will need to directly prove an SQ lower bound for the search problem.

Consider the following adversary for the search problem. The adversary will let $X = \mathbf{P}_{\mathbf{v}}^A$ for $\mathbf{v} \sim U(\mathbb{S}^{n-1})$ be the input distribution, and whenever possible, the adversary will answer a query with $\mathbf{E}_{\mathbf{x} \sim \mathcal{N}_n}[f(\mathbf{x})]$. Given that the algorithm asks less than $2^{n^{\Omega(1)}}$ queries, as we have shown in the proof of Theorem 1.5, with $1 - o(1)$ probability, the adversary can always answer $\mathbf{E}_{\mathbf{x} \sim \mathcal{N}_n}[f(\mathbf{x})]$. In such case, the algorithm will be left with $1 - o(1)$ probability mess over $\mathbf{v} \sim U(\mathbb{S}^{n-1})$ that are equally likely.

Then we argue that no hypothesis can be close to more than $2^{-\Omega(n)}$ probability mass. This can be done by upper bounding the surface area of a spherical cap on a $n$-dimensional sphere, where the sphere is unit radius and the polar angle of the cap is a sufficiently small constant $\Phi$. Note that the surface area of such a cap is $\Theta\left(I_{\sin^2 \Phi}\left(\frac{n-1}{2}, \frac{1}{2}\right)\right) S_{n-1}(\mathbb{S}^{n-1})$, where $I$ is the incomplete beta function and $S_{n-1}(\mathbb{S}^{n-1})$ is the surface area of the $n$ dimensional unit sphere. Thus, it suffices to show that $I_{\sin^2 \Phi}\left(\frac{n-1}{2}, \frac{1}{2}\right) = 2^{-\Omega(n)}$. Notice that, by its definition, we have

$$
\begin{aligned}
I_{\sin^2 \Phi}\left(\frac{n-1}{2}, \frac{1}{2}\right) &= \frac{\int_0^{\sin^2 \Phi} t^{(n-3)/2}(1-t)^{-1/2}dt}{B\left(\frac{n-1}{2}, \frac{1}{2}\right)} \\
&= \frac{\int_0^{\sin^2 \Phi} t^{(n-3)/2}(1-t)^{-1/2}dt}{B\left(\frac{n-1}{2}, 1\right)} \frac{B\left(\frac{n-1}{2}, 1\right)}{B\left(\frac{n-1}{2}, \frac{1}{2}\right)} \\
&\leq O(1)\frac{\int_0^{\sin^2 \Phi} t^{(n-3)/2}(1-t)^{0}dt}{B\left(\frac{n-1}{2}, 1\right)} \\
&= O(1)I_{\sin^2 \Phi}\left(\frac{n-1}{2}, 1\right) = O(1)(\sin^2 \Phi)^{(n-1)/2} = 2^{-\Omega(n)}.
\end{aligned}
$$

Given that no hypothesis can be close to more than $2^{-\Omega(n)}$ probability mass, the only way to have any constant probability of success would be to return $2^{\Omega(n)}$ many hypotheses. This completes the proof. $\qquad \square$

## C.2 Proof of Lemma 1.9

*Proof.* We will take $E$ to be the distribution with density $E(t) = \mathcal{N}(t) + \mathbb{1}(t \in [-1, 1])p(t)$, where $p$ is a polynomial function that we truncate between $[-1, 1]$. In order to satisfy our requirements, it will suffice to have $|p(t)| \leq 1/10$ for all $t \in [-1, 1]$ and for each integer $0 \leq i \leq d$,

$$
\mathbf{E}_{t \sim A}[t^i] = (1-\alpha_d)\mathbf{E}_{t \sim E}[t^i] = (1-\alpha_d)\mathbf{E}_{t \sim \mathcal{N}}[t^i] + (1-\alpha_d)\int_{-1}^{1} P(t)t^i dt = \mathbf{E}_{t \sim \mathcal{N}}[t^i].
$$

The second requirement is equivalently stated as follows: for each such $i$,

$$\int_{-1}^{1} P(t)t^i dt = \frac{\alpha_d}{1 - \alpha_d} \mathbf{E}_{t \sim \mathcal{N}}[t^i] \ .$$

In order to satisfy these requirements, we need the following fact from [DK23].

**Fact C.1** (Lemma 8.18 in [DK23]). *Let $C > 0$ and $m \in \mathbb{Z}_+$. For any $a_0, a_1, \cdots, a_m \in \mathbb{R}$, there exists a unique degree at most $m$ polynomial $p : \mathbb{R} \to \mathbb{R}$ such that for each integer $0 \le t \le k$ we have that*

$$\int_{-C}^{C} p(x)x^t dx = a_t \ .$$

*Furthermore, for each $x \in [-C, C]$ we have that $|p(x)| \le O_m(\max_{o \le t \le m} a_t C^{-t-1})$.*

We apply the fact and take $C = 1$. This implies that there is such a polynomial with $|p(x)| \le 1/10$ for sufficiently small $\alpha_d$ depending only on $d$. $\qquad\square$

### C.3   Proof of Theorem 1.10

We provide a more detailed statement of Theorem 1.10 here.

**Theorem C.2** (SQ Lower Bound for AC Detection). *There exists a function $f : (0, 1/2) \to \mathbb{N}$ that $\lim_{\alpha \to 0} f(\alpha) = \infty$ and satisfies the following. For any sufficiently small $\alpha \in (0, 1/2)$, any SQ algorithm that has access to a distribution that is either (i) a standard Gaussian; or (ii) a distribution that has at least $\alpha$ probability mass in a $(n-1)$-dimensional subspace $V \subset \mathbb{R}^n$, and distinguishes the two cases with success probability at least $2/3$, either requires a query with error at most $O_\alpha(n^{-f(\alpha)})$, or uses at least $2^{n^{\Omega(1)}}$ many queries.*

*Proof of Theorem C.2.* This is a direct application of Theorem 1.5. We will let the one-dimensional moment-matching distribution be the distribution in Lemma 1.9, where we will take $d$ only depends on $\alpha$ to be the largest integer such that $\alpha_d$ in Lemma 1.9 satisfies $\alpha_d \ge \alpha$. Notice that $d \to \infty$ as $\alpha \to 0$. Taking $f(\alpha) = d/32$, it follows that any algorithm distinguishing between

- A standard Gaussian; and

- The distribution $\mathbf{P}_{\mathbf{v}}^A$ for $\mathbf{v} \sim U(\mathbb{S}^{n-1})$, where $A$ is the moment-matching distribution in Lemma 1.9,

with at least $2/3$ probability must require either a query of tolerance at most $O_d(n^{-d/32}) = O_\alpha(n^{-f(\alpha)})$ or $2^{n^{\Omega(1)}}$ many queries. Notice that the distribution $\mathbf{P}_{\mathbf{v}}^A$ has at least $\alpha$ probability mass resides inside the orthogonal complement of $\text{span}(\mathbf{v})$, which is a $(n-1)$-dimensional subspace. Therefore, any SQ algorithm for solving the AC detection solves the hypothesis testing problem above. This completes the proof. $\qquad\square$

### C.4   Proof of Theorem 1.13

Let $G'_{s,\theta}$ be the probability measure obtained by rescaling $G'_{s,\theta}$ such that the total measure is one. We first show the following fact.

**Fact C.3.** *For any polynomial $p$ of degree at most $k$ that $\mathbf{E}_{t \sim \mathcal{N}(0,1)}[p(t)^2] = 1$, $s$ that is at most a sufficiently small universal constant, $|\mathbf{E}_{t \sim G'_{s,\theta}}[p(t)] - \mathbf{E}_{t \sim \mathcal{N}(0,1)}[p(t)]| = k!2^{O(k)} \exp(-\Omega(1/s^2))$.*

*Proof.* Using Fact 1.12, we have that the total measure of $G_{s,\theta}$ is $1 \pm \exp(-\Omega(1/s^2))$. Therefore, for the rescaled $G'_{s,\theta}$, for any $k \in \mathbb{N}$, $s > 0$ and all $\theta \in \mathbb{R}$,

$$|\mathbf{E}_{t \sim \mathcal{N}(0,1)}[t^k] - \mathbf{E}_{t \sim G'_{s,\theta}}[t^k]| = k! \exp(-\Omega(1/s^2)) \ .$$

Then using the definition of Hermite polynomial,

$$\mathbf{E}_{t\sim\mathcal{N}(0,1)}[h_k(t)] - \mathbf{E}_{t\sim G'_{s,\theta}}[h_k(t)] = \frac{1}{\sqrt{k!}}\sum_{t=0}^{\lfloor k/2\rfloor}\frac{k!}{2^t t!(k-2t)!}\left(\mathbf{E}_{t\sim\mathcal{N}(0,1)}[t^{k-2t}] - \mathbf{E}_{t\sim G'_{s,\theta}}[t^{k-2t}]\right)$$

$$\leq \left(k!\exp(-\Omega(1/s^2))\right)\sum_{t=0}^{\lfloor k/2\rfloor}\frac{\sqrt{k!}}{2^t t!(k-2t)!}t^{k-2t}\ .$$

Notice that the denominator is minimized when $t = k/2 - O(\sqrt{k})$. Then it follows that the sum is at most $k!2^{O(k)}\exp(-\Omega(1/s^2))$. Now let $p(t) = \sum_{i=0}^{k}\mathbf{w}_i h_i(t)$. Since $\mathbf{E}_{t\sim\mathcal{N}(0,1)}[p(t)^2] = 1$, it must be $\|\mathbf{w}\|_2 = 1$ and it follows that $\|\mathbf{w}\|_1 \leq \sqrt{k}$. Therefore, we have

$$\mathbf{E}_{t\sim\mathcal{N}(0,1)}[p(t)] - \mathbf{E}_{t\sim G'_{s,\theta}}[p(t)] \leq \sum_{i=0}^{k}|\mathbf{w}_i||\mathbf{E}_{t\sim\mathcal{N}(0,1)}[h_k(t)] - \mathbf{E}_{t\sim G'_{s,\theta}}[h_k(t)]|$$

$$= k!2^{O(k)}\exp(-\Omega(1/s^2))\ .$$

$\square$

Now given Fact C.3, we can apply our main result Theorem 1.5. We will consider the following distribution distinguishing problem. In both the null hypothesis case and the alternative hypothesis case, the algorithm is given a joint distribution $D$ of $(\mathbf{x}, y)$ over $\mathbb{R}^n \times \mathbb{R}$. In the null hypothesis case, we have $\mathbf{x} \sim \mathcal{N}(0, \mathbf{I}_n)$ and $y \sim U([-1, +1])$ independently. While in the alternative hypothesis case, we have $\mathbf{x} \sim \mathcal{N}(0, \mathbf{I}_n)$ and $y = \cos(2\pi(\delta\langle\mathbf{w}, \mathbf{x}\rangle + \zeta))$ with noise $\zeta \sim \mathcal{N}(0, \sigma^2)$ as in the definition of learning periodic function. Notice that any SQ algorithm that can always returns a hypothesis $h$ such that $\mathbf{E}_{(\mathbf{x},y)\sim D}[(h(\mathbf{x}) - y)^2] = o(1)$ can also be easily used to distinguish the two cases. Therefore to lower bound such SQ algorithms, it suffices for us to give an SQ lower bound for this distribution distinguishing problem.

Notice that in the alternative hypothesis, the distribution of $\mathbf{x}$ conditioned on any value of $y$ is the hidden direction distribution $\mathbf{P}_{\mathbf{w}}^A$ where $A$ is $\int_{-\infty}^{\infty} f_y(\zeta)\frac{1}{2}\left(G'_{1/\delta,\frac{\arccos(y)}{2\pi}-\zeta} + G'_{1/\delta,\frac{2\pi-\arccos(y)}{2\pi}-\zeta}\right)d\zeta$ and $f_y : \mathbb{R} \to \mathbb{R}$ is the PDF function of distribution of $\zeta$ conditioned on y. Notice that this is a mixture of $G_{s,\theta}$ with $s = 1/\delta$ and different $\theta$. Thus applying Theorem 1.5 and Fact C.3 yields that any SQ algorithm for solving the distinguishing problem, either requires a query of error at most $O_k(n^{-((1-\lambda)/8-\beta)k}) + k!2^{O(k)}\exp(-\Omega(1/s^2))$, or at least $2^{n^{\Omega(\beta)}}$ many queries for $\beta > 0$. Since $k$ will have dependence on $n$, we will need to calculate the constant factor in $O_k(n^{-((1-\lambda)/8-\beta)k})$ which depends on $k$. According to Proposition 3.1, plug in the factor gives $\sqrt{k!}n^{-((1-\lambda)/8-\beta)k} + k!2^{O(k)}\exp(-\Omega(1/s^2))$.

For convenience, let $(1-\lambda)/8 - \beta = \gamma$. It only remains to choose the value of $k$ and $\gamma$ so that $k \leq n^\lambda$, $\gamma < (1-\lambda)/8$ and $\sqrt{k!}n^{-\gamma k} + k!2^{O(k)}\exp(-\Omega(\delta^2))$ is minimized. We will chose $k = n^{c''}$ and $\gamma = c''$ for $c' < c'' < \min(2c, 1/10)$. Then the error tolerance here is $\sqrt{k!}n^{-\gamma k} + k!2^{O(k)}\exp(-\Omega(\delta^2)) \leq \exp(-k) + \exp(O(c''(\log n)n^{c''}) + O(n^{c''}) - \Omega(n^{2c})) = \exp(-\Omega(n^{c''})) + \exp(-\Omega(n^{2c})) = \exp(-n^{c'})$. The number of queries here is $2^{n^{\Omega(\beta)}} = 2^{n^{\Omega((1-\lambda)/8-\gamma)}} = 2^{n^{\Omega((1-c'')/8-c')}} = 2^{n^{\Omega(1)}}$. This completes the proof.

## C.5 SQ Lower Bounds as Information-Computation Tradeoffs

We note that both aforementioned results can be viewed as evidence of information-computation tradeoffs for the application problems we discussed.

For the problem of list-decodable Gaussian mean estimation, the information-theoretically optimal error is $\Theta(\log^{1/2}(1/\alpha))$ and is achievable with $\text{poly}(n/\alpha)$ many samples ([DKS18]; see, e.g., Corollary 5.9 and Proposition 5.11 of [DK23]). The best known algorithm for this problem achieves $\ell_2$-error guarantee $O(\alpha^{-1/d})$ using sample complexity and run time $(n/\alpha)^{O(d)}$ ([DKS18], see Theorem 6.12 of [DK23]). Notice that in order to achieve error guarantee even sub-polynomial

in $1/\alpha$, the above algorithm will need super-polynomial runtime and sample complexity. Informally speaking, Theorem 1.8 shows that no SQ algorithm can perform list-decodable mean estimation with a sub-exponential in $n^{\Omega(1)}$ many queries, unless using queries of very small tolerance — that would require at least super-polynomially many samples to simulate. Therefore, it can be viewed as evidence supporting an inherent tradeoff between robustness and time/sample complexity for this problem.

For the AC detection hypothesis testing problem, the information-theoretically optimal sample complexity is $O(n/\alpha)$. To see this, note that if the input distribution is a standard Gaussian, then any $n$ samples will almost surely be linearly independent. On the other hand, suppose that the input distribution has $\alpha$ probability mass in a subspace. Then with $O(n/\alpha)$ many samples, with high probability, there will be a subset of $n$ samples all coming from that subspace, which cannot be linearly independent. However, our SQ lower bound suggests that no efficient algorithm can solve the problem with even $n^{\omega(1)}$ samples where the $\omega(1)$ is w.r.t. $\alpha \to 0$. This suggests an inherent tradeoff between the sample complexity and time complexity of the problem.

For the problem of learning periodic function, the SQ lower bound given by our result will be larger than the algorithmic upper bound in [SZB21](with sample complexity $O(n)$ and run-time $2^{O(n)}$). However, the algorithms in [SZB21] are based on LLL lattice-basis-reduction which is not captured by the SQ framework, therefore this does not contradict our SQ lower bound result.