# OpenReview forum: "SQ Lower Bounds for Non-Gaussian Component Analysis with Weaker Assumptions"
_NeurIPS.cc/2023/Conference — NeurIPS 2023 poster_

### Official Review · Reviewer_g3gW · 2023-06-30

**Soundness:** 2 fair
**Presentation:** 1 poor
**Contribution:** 2 fair
**Rating:** 4
**Confidence:** 2

**Summary:**

This paper provides an SQ lower bound for the problem of distinguishing the standard Gaussian distribution from a distribution with a single non-Gaussian component, under relaxed assumptions compared to what was previously known. In particular, in prior work, similar SQ lower bounds were established in the case where the non-Gaussian component corresponded to any single-dimensional distribution $A$ whose low order moments match those of a standard Gaussian and the ($\chi^2$) distance between $A$ and the standard Gaussian is bounded. In this work, the bounded ($\chi^2$) distance assumption is removed.

As consequences of the main result, two additional SQ lower bounds are provided, one for List-Decodable Mean estimation and one for the problem of anti-concentration detection.

The approach is simple and consists of direct probabilistic bounds for the difference between the value of a statistical query under the Gaussian and the corresponding value under a distribution with a non-Gaussian component in a uniformly random direction. The bound is shown via Fourier analysis over the Hermite basis.

**Strengths:**

The problem is clear and well motivated. The claimed result is a strict improvement over prior work with some further applications.

**Weaknesses:**

The paper has several important weaknesses.

1. The main result seems to contradict prior work (which is not cited in the paper), which raises serious doubts about its correctness. In particular, in Theorem 1.3 of [DK22] it is shown that non-Gaussian components that are far from the unit normal are detectable (even under the moment matching assumption). See also the discussion on page 2, paragraph "Motivation of This Work" of [DK22].

[DK22]:  Diakonikolas, Ilias, and Daniel Kane. "Non-gaussian component analysis via lattice basis reduction." Conference on Learning Theory. PMLR, 2022.

2. The presentation of related work contains inaccuracies. In particular, within the paragraph in lines 66-74, it is implied that all of the listed papers make use of results on the hardness of non-Gaussian component analysis (NGCA) to gain results for other problems. However, for example, [GGJ+20] does not even mention the problem of NGCA, but makes use of some appropriate combinatorial quantity (called statistical dimension) to arrive to their main result.

3. The text in the introduction is sloppy and overly casual and section 3 is too technical. For example, lines 52-57 could use some polish and lines 148-153 give the impression that there are no further technical challenges, so the following paragraph becomes confusing.

-- The authors addressed my main concern that their results seemed to contradict prior work.

**Questions:**

1. My main question concerns the correctness of the main result. How does this result compare to Theorem 1.3 of [DK22]? Such a discussion should, in any case, be a part of the paper.

2. Section 3 is quite dense and cannot be easily parsed, so it might be beneficial to substitute the proofs with proof sketches that contain less quantitative information (the full proofs can be moved to the Appendix).

**Limitations:**

The existence of an LLL-based algorithm for the problem in question [DK22] should be discussed thoroughly.

[DK22]:  Diakonikolas, Ilias, and Daniel Kane. "Non-gaussian component analysis via lattice basis reduction." Conference on Learning Theory. PMLR, 2022.

---

> ### Author Rebuttal · Authors · 2023-08-10
>
> We thank the reviewer for their effort and feedback on our work. We would like to address the following questions/comments.
>
> 1. Regarding the reviewer’s point that the main result seems to contradict the algorithm in prior work [ZSWB22] and [DK22].
>
> If we take the one-dimensional distribution $A$ in the NGCA to be a distribution with finite support and matching the first $k$ moments with the standard Gaussian, where $k$ is at least a sufficiently large integer, then the SQ lower bound given by our result will be larger than the algorithmic upper bound in [ZSWB22] and [DK22]. However, as also pointed out by reviewer nJue, the algorithms in [ZSWB22] and [DK22] are based on LLL lattice-basis-reduction which is not captured by the SQ framework, so this does not contradict our SQ lower bound result.
>
> Another classical example is the exponential SQ-hardness of learning Parities, when a polynomial-time algorithm (based on Gaussian elimination) exists for the problem. Interestingly, the same situation holds for low-degree polynomial tests and SoS bounds (i.e., these restricted models of computation do not capture LLL or Gaussian elimination).
>
> We would also like to point out that the above does not imply that any NGCA family of instances with infinite chi-square distance can be solved efficiently. Importantly, LLL-based algorithms only work in the restricted setting that the support of the one-dimensional distribution A is discrete/near-discrete. For example, if the one-dimensional distribution $A$ is a mixture of a discrete distribution and a continuous distribution (as is the case  in the Anti-concentration Detection problem considered in our paper), linear-algebraic algorithms will not work although the distribution has infinite chi-square distance and the problem is believed to be hard for all efficient algorithms.
>
> 2. Regarding the review's point about presentation of Related Work:
>
> Within the paragraph in lines 66-74, we have provided an extensive — but not exhaustive –  list of 12 papers using NGCA to establish SQ lower bounds. The reviewer points out that one of these papers, namely  [GGJ+20], does not explicitly use the NGCA problem. Upon inspection, one can see that the CSQ hard instance in [GGJ+20] uses a construction very similar to the concurrent work [DKKZ20] (which shows a quantitatively stronger lower bound for the same problem). The lower bound construction of [DKKZ20] explicitly relies on NGCA. This connection is discussed in the second to last paragraph of the “Our Techniques” section of [DKKZ20].
>
> Reference:\
> [DK22] I. Diakonikolas and D. M. Kane. Non-Gaussian Component Analysis via Lattice Basis Reduction. In Conference on Learning Theory, COLT 2022, volume 178 of Proceedings of Machine Learning Research, pages 4535-4547. PMLR, 2022.\
> [GGJ+20] S. Goel, A. Gollakota, Z. Jin, S. Karmalkar, and A. R. Klivans. Superpolynomial lower bounds for learning one-layer neural networks using gradient descent. In Proceedings of the 37th International Conference on Machine Learning, ICML 2020, volume 119 of Proceedings of Machine Learning Research, pages 3587–3596. PMLR, 2020.\
> [ZSWB22] I. Zadik, M. J. Song, A. S. Wein and J. Bruna. Lattice-Based Methods Surpass Sum-of-Squares in Clustering. In Conference on Learning Theory, COLT 2022, volume 178 of Proceedings of Machine Learning Research, pages 1247-1248. PMLR, 2022.

---

> > ### Comment · Reviewer_g3gW · 2023-08-11
> >
> > I thank the authors for responding to my comments and questions and for resolving a part of my concerns.
> >
> > 1. While the authors' response alleviated my concern that their work contradicts existing results in the literature, I believe that comparing their results to [ZSWB22] and [DK22] should be a central theme in their paper and I still find it surprising (and confusing) that they did not even mention these results in the first version of the paper. In particular, the existence of these algorithms should have been thoroughly discussed as a limitation of their work.
> > In view of the results in [ZSWB22] and [DK22], the provided SQ lower bounds seem less appealing, since a non-SQ algorithm is already known for (a restricted version of) the setting and therefore the SQ framework might not be the correct lens to study the problem; in contrast, for other problems in learning theory, SQ lower bounds are often accompanied by similar cryptographic hardness results that apply to all algorithms. That said, from a purely theoretical perspective, it is interesting that there is a separation between SQ and non-SQ algorithms in the setting. Also, I appreciate the fact that LLL-based algorithms are brittle and SQ lower bounds might be useful to rule out more robust approaches.
> >
> > 2. I strongly believe that the paragraph in lines 66-74 is confusing and even misleading. While I only provided one example in my first response, the first sentence of the paragraph ("The SQ-hardness of NGCA can be used to obtain similar hardness for a number of well-studied learning problems that superficially appear very different.") does not do justice to at least the following papers: [DKKZ'20], [GGK'20], [CLL'22], [GGJ+20]. The authors claim that "the SQ-hardness of NGCA can be used to obtain similar hardness for a number of well-studied learning problems", which suggests that because NGCA is hard, other problems are also hard, implying that some reduction-based approach would work. However, for each of the aforementioned papers, this is not true. Instead, some of these papers use some tools from the foundational paper of [DKS'17] ([GGJ+20] does not -- they do not even cite [DKS'17]) , but these tools are not tied to NGCA, but to the SQ-framework in general (see, e.g., Fact 3.4 from [CLL'22]). The section "Our Techniques" from [DKKZ'20] (which the authors pointed to in their rebuttal) only acknowledges this fact: "To achieve this, we build on an idea introduced in [DKS17]." It is not surprising that ideas used to prove one kind of SQ lower bound can be used to prove another SQ lower bound, and if it is, it only implies that the specific ideas in question are important and not necessarily that any results on the problem of NGCA are. A working (but still potentially confusing) alternative to the first sentence of the paragraph 66-74 might be for example: "Proofs of SQ-hardness of NGCA in foundational prior work [DKS'17] have constituted technical and/or conceptual points of reference for proofs of hardness of a variety of well-studied learning problems."
> >
> > **References:**
> >
> > [DK22] I. Diakonikolas and D. M. Kane. Non-Gaussian Component Analysis via Lattice Basis Reduction. In Conference on Learning Theory, COLT 2022, volume 178 of Proceedings of Machine Learning Research, pages 4535-4547. PMLR, 2022.
> >
> > [ZSWB22] I. Zadik, M. J. Song, A. S. Wein and J. Bruna. Lattice-Based Methods Surpass Sum-of-Squares in Clustering. In Conference on Learning Theory, COLT 2022, volume 178 of Proceedings of Machine Learning Research, pages 1247-1248. PMLR, 2022.
> >
> > \+ Reverences in the paper

---

> > > ### Author Response · Authors · 2023-08-12
> > >
> > > We thank the reviewer for acknowledging that their prior concern (regarding correctness issues of our main SQ lower bound) is now resolved. We remind the readers that the context and our response on this point was also provided in the overall response to all reviewers (first paragraph after the quoted question by the reviewer).
> > >
> > > Below we respond to the additional points made by the reviewer:
> > > - (Significance of NGCA SQ Lower Bounds in Our More General Setting) In addition to its theoretical significance, our general SQ lower bound for NGCA (without the chi-squared bound assumption) has applications to concrete learning problems that are believed to be computationally hard. In the paper, we provided two applications – one for list-decodable mean estimation and one for the anti-concentration detection problem.
> > >     - For the former problem, we prove (Theorem 1.8) an SQ lower bound for a broader set of parameters compared to prior work (corresponding to small values of the parameter $\alpha$). Notably, the chi-squared distance for these instances is very large but finite, so that an application of the [DKS17] SQ lower bound for NGCA gives vacuous results.
> > >     - For the latter problem, we establish the first super-polynomial SQ hardness (Theorem 1.10). For the corresponding SQ-hard instances, the chi-squared distance is infinity.
> > >     - In both of these applications,  the SQ-hard instances cannot be learned efficiently via LLL-type algorithms. Please see the overall response to all reviewers (second paragraph after the quoted question by the reviewer).
> > >
> > > - (Discrete Moment-matching Distribution) Even for the very special case where the one-dimensional distribution $A$ is (nearly) discrete, we believe that our SQ lower bound is conceptually interesting (a view shared by other reviewers). To reiterate, our SQ lower bound for this special case implies that LLL-based-algorithms are not captured in the SQ framework. This is a novel and interesting limitation of SQ algorithms. It was previously known that this limitation is shared by two other prominent restricted families of algorithms (namely, SoS algorithms and low-degree polynomial tests); it was unknown whether SQ algorithms have this limitation. As a corollary, we now know of two “exceptional” algorithms that are not efficiently implementable in these models: Gaussian elimination (for learning parities) — this is a classical result — and LLL-based algorithms (as follows from our work for the class of SQ algorithms).
> > > - In summary, we respectfully disagree with the reviewer’s subjective point of view, namely that “SQ lower bounds seem less appealing” in this setting. Moreover, we consider the statement “SQ lower bounds are *often* accompanied by similar cryptographic hardness results” inaccurate. While there are a few concrete problems where SQ lower bound constructions have led to similar crypto hardness, this is not true in most of the cases – hence, the term “often” is factually incorrect.
> > > - (Presentation) Here we address a couple of points on the presentation of our work raised by the reviewer:
> > >     - (Remark on LLL-based algorithms) As we noted in our initial response, we will add a remark in the revised version of our paper explaining the connection to recent LLL-based algorithms and the associated implications. We respectfully disagree with the reviewer’s statement that the comparison should be “a central theme” in our paper, because the corresponding instances (for discrete $A$) are a very special case of the instances we establish SQ-hardness for and not a central focus/application of our general theorem.
> > >     - (Paragraph in lines 66-74) We are happy to revise this paragraph, as it seems to have been confusing to the reviewer. The punchline of this paragraph is that the papers we have listed share the following property: **The SQ-hard instances in these papers are (SQ-hard) instances of NGCA for a specific choice of the moment-matching distribution $A$.** In other words, the SQ-hardness results in these works are obtained by applying a generic NGCA SQ-hardness result (either from [DKS17] or a  natural generalization thereof). To achieve this, for each of these works, the authors construct a distribution $A$ that satisfies Condition 1.4 (for problem-specific values of $m$ and $\chi^2(A, N(0, 1))$) such that the corresponding $P_{\bf v}^{A}$ (Definition 1.2) belongs to the class of distributions being learned. In summary, all these SQ-hardness results are literally reductions from instances of the corresponding learning problem to *specific* instances of NGCA. The reviewer is referred to Section 8.4 (third paragraph) of the recent book [DK23] on algorithmic robust statistics surveying this line of work.

---

> > > > ### Author Response · Authors · 2023-08-12
> > > >
> > > > Reference:\
> > > > [DK23] I. Diakonikolas and D. M. Kane. Algorithmic High-Dimensional Robust Statistics. Cambridge University Press, 2023.\
> > > > [DKS17] I. Diakonikolas, D. M. Kane, and A. Stewart. Statistical query lower bounds for robust estimation of high-dimensional gaussians and gaussian mixtures. In 58th IEEE Annual Symposium on Foundations of Computer Science, FOCS 2017, pages 73–84, 2017.

---

> > > > ### Comment · Reviewer_g3gW · 2023-08-12
> > > >
> > > > Thank you for another detailed response and for pointing out that the term "often" in my previous response was not used accurately. I still believe that the existence of the LLL-based algorithm is a significant limitation of the paper that should be discussed thoroughly and that certain claims in the paper need to be further justified and/or clarified. In light of the authors' rebuttal and further responses, I will increase my score accordingly, but I will also decrease my confidence.
> > > >
> > > > It would be appreciated if the authors could instantiate (at least in high level) their claim that: "The SQ-hard instances in these papers are (SQ-hard) instances ...  literally reductions from instances of the corresponding learning problem to specific instances of NGCA." for the lower bound provided in [DKKZ'20] and/or [GGJ+20] (i.e., which is the constructed distribution A and which is the corresponding $P_v^A$?).

---

> > > > ### Comment · Reviewer_oUue · 2023-08-14
> > > >
> > > > A comment about the LLL vs. SQ discussion: I agree with the opinion that a discussion on the relation to the LLL algorithm should be added to the paper. I would like to add to this that, as far as I can tell, the observation that LLL can surpass SQ lower bounds is *not* novel and has previously appeared in the literature in the form of a published paper (Ding, Jingqiu; Hua, Yiding; "SQ Lower Bounds for Random Sparse Planted Vector Problem", ALT'23). This should be acknowledged in the discussion.

---

> > > > > ### Author Response · Authors · 2023-08-15
> > > > >
> > > > > We thank reviewer oUue for their remark. We reiterate, as we mentioned in our initial response, that we will add a paragraph explaining this connection in detail. We will also cite the ALT’23 paper mentioned by the reviewer.
> > > > >
> > > > > Regarding the referenced ALT’23 paper: We agree with the reviewer that a separation between SQ and LLL follows from that paper. In more detail, this paper shows a quadratic separation between the information-theoretic sample complexity (which is $\Theta(d)$) and the computational sample complexity in the SQ model (which is $\Theta(d^2)$).  The reason is that the paper establishes an SQ lower bound for the very special case that the distribution $A$ is discrete supported on three points (namely,  $\\{-1, 0, 1\\}$). Such a distribution can match up to $k=3$ moments with $N(0, 1)$ which in turn leads to a quadratic sample lower bound. In contrast, our general SQ lower bound can be applied for any discrete distribution (of any support size) that matches $k$ moments (for any $k$ that is at most $O(d)$). (We note that there exists a discrete $A$ of support $k$ that matches $2k-1$ moments with $N(0,1)$, via Gaussian quadrature techniques. See e.g., Lemma 4.3 in the arxiv version of [DKS17].) As a result, the corollary obtained from our general SQ result (for the special case of discrete distributions) is a super-polynomial separation between the information-theoretic sample complexity and the computational sample complexity (which is $\Theta(d^{\Omega(k)})$ for some $k = \omega(1)$).

---

### Official Review · Reviewer_oUue · 2023-07-01

**Soundness:** 3 good
**Presentation:** 3 good
**Contribution:** 3 good
**Rating:** 7
**Confidence:** 3

**Summary:**

This work studies non-Gaussian component (NGCA) analysis in the context of the statistical query model. In NGCA the task is to distinguish a standard multi-variate Gaussian distribution from a distribution that is standard Gaussian in all but a random direction $w$ and equal to a one-dimensional distribution $A$ along $w$.

Previously, it was known that this problem is hard in the SQ model when
1. $A$ matches many moments with the (one-dimensional) standard Gaussian,
2. the $\chi^2$-divergence between $N(0,1)$ and $A$ is finite.

Also, the quality of the lower bound depends on the $\chi^2$-divergence: If this is very large with respect to other problem parameters (e.g., the dimension and the number of moments matched), the lower bound can potentially be weaker.

In their work, the authors show that Assumption (2) above is not necessary for the hardness result to hold. They use this to obtain an improved result for list-decodable mean estimation (when the inliers are Gaussian with identity-covariance). They further show SQ hardness of detecting whether a distribution has a constant fraction of its probability mass in a lower-dimensional subspace.

On a technical level, they are able to remove the assumption on the $\chi^2$-divergence since they do not show hardness based on the SQ dimension (a widely used notion which implies SQ hardness), but rather show directly that every statistical query on the two distributions must receive the same answer up to some small error. They show this by carefully analyzing the Fourier moments of both distributions.

**Strengths:**

In my eyes, the main strength of the paper is that it allows for more flexibility when using the NGCA framework. Second, it also makes it easier to apply. Since this framework has found numerous applications, I believe that many researchers will appreciate this result. On a conceptual level, it is pleasing to see that the $\chi^2$-divergence condition is indeed just a technicality (and the "true hardness" comes from the moment-matching condition).

I believe the result can lead to:
1. improved lower bounds (since the $\chi^2$-divergence did affect the quality of the lower bound),
2. and simple(r) constructions (since we now have more flexibility).

The authors gave one example for each of the two in their paper.

The paper is generally well-written. The authors give a nice and easy-to-follow overview of the main result in Section 1.2. Their solution is very natural. The actual proofs are more technical, but this is shared with other SQ lower bounds using the NGCA framework.

**Weaknesses:**

Minor: While the quantitative improvement for the list-decoding lower bound is non-trivial, it appears only in a very restricted regime of parameters, when the fraction of inliers goes to 0 when the dimension grows.


**Questions:**

In some of the existing SQ lower bounds using the NGCA framework it was very useful if the moments of $A$ only need to *approximately* match those of a standard Gaussian (e.g., the lower bounds for learning halfspaces under Massart noise in the distribution-independent setting). Are your techniques able to capture this scenario as well? If this is the case, it would be very helpful to include it in the main theorems since it would make them even easier to apply.

**Limitations:**

The authors give a fairly definitive answer to the question they studied.

---

> ### Author Rebuttal · Authors · 2023-08-10
>
> We thank the reviewer for their effort and positive assessment of our work. We would like to address the following questions/comments.
>
> 1. Regarding the reviewer’s point:
> “While the quantitative improvement for the list-decoding lower bound is non-trivial, it appears only in a very restricted regime of parameters, when the fraction of inliers goes to $0$ when the dimension grows.”
>
> We point out that this setting of “small $\alpha$” (e.g., subconstant in the dimension) is actually of interest in some applications, including in mean estimation. A concrete example in a related crowdsourcing setting is the COLT 2018 paper by Meister and Valiant [MV18] dealing with this  “small $\alpha$” parameter regime.
>
> 2. Regarding the reviewer’s point:
> “Does the technique here capture the scenario that $A$ only approximately matches moments with the Gaussian?”
>
> Yes, our technique captures the approximate moment-matching scenario. The only difference in the statement and proof would be an extra term due the approximate matching-moment in all the calculations.
>
> Reference:\
> [MV18] M. Meister and G. Valiant. A Data Prism: Semi-verified learning in the small-alpha regime. In Conference on Learning Theory, COLT 2018, volume 75 of Proceedings of Machine Learning Research, pages 1530-1546. PMLR, 2018.

---

> > ### Comment · Reviewer_oUue · 2023-08-14
> >
> > Thank you for your comments. Regarding your second point: That's great to hear. I would again encourage the authors to include this as a formal result in their submission (in the appendix is fine if they wish not to clutter the main text).

---

### Official Review · Reviewer_YVEY · 2023-07-03

**Soundness:** 3 good
**Presentation:** 3 good
**Contribution:** 4 excellent
**Rating:** 7
**Confidence:** 2

**Summary:**

The paper considers the SQ-hardness of non-Gaussian component analysis. The main result of the paper is a statement of the hardness without an assumption required by results stated in previous works: finite chi-squared distance between the non-gaussian distribution and the standard normal. Two applications are discussed. The bulk of the paper is focused on proof techniques, which makes use of Fourier transforms and Hermite polynomials to bound expected values of query functions.

**Strengths:**

The main result of the paper by itself is interesting enough to be the greatest strength of the paper.

Originality. While the topic of hypothesis testing of NGCA in the Statistical Query (SQ) model is not new, and it has been known that the problem itself is SQ hard, this paper removes one of the main assumptions required for results in previous works to hold. From this perspective, the paper provides original contributions to the field.

Quality. The quality of the paper is quite nice. The paper does a good job in describing the problem formulation, downfalls of past approaches, and definitions of the various technical tools required to establish the proof of the main result.

Clarity. The paper is well written and evidently polished. The main contributions of the paper are well stated and clear. The proof sections in the last few pages of the paper may need a bit more revision to be better readable.

Significance. The main results of the paper are widely applicable to various areas of machine learning theory.

**Weaknesses:**

Overall, the paper is nicely written. Some (minor) comments on weaknesses of the paper are:

1. It is stated in the paper that the main result is "near-optimal". Can some elaborations be made on what is the "optimal" result, and, what are the technical difficulties causing the discrepancy between the results in this paper and the "optimal" result?

2. Discrepancy in order of Theorem 1.8: the results in the paper give a tolerance requirement of the order "$Ω(d)^{−k/32}$" on $d$ while Fact 1.6. has the order "$Ω(d)^{-(k+1)(1/4-c/2)}$". In cases where $c$ is closer to $1/2$ it appears that the previous result is tighter, while the opposite is true when $c$ is closer to $0$. What is causing the discrepancy and why is this not reflected in the original comparison in line 62, i.e., is the order of magnitude listed in the equation on line 62-63 inaccurate?

**Questions:**

Some additional comments and questions:

1. Possible typo on Line 289: "i-th" might have been a typo of "k-th".

2. Lemma 3.10 requires that $d$ "is at least a sufficiently large universal constant." How large does $d$ need to be in order for the results in the paper to be applicable?

**Limitations:**

There are limited discussions on limitations available in the paper.

---

> ### Author Rebuttal · Authors · 2023-08-10
>
> We thank the reviewer for their effort and positive assessment of our work. We would like to address the following questions/comments.
>
> 1. Regarding the reviewer’s point:
> “It is stated in the paper that the main result is "near-optimal". Can some elaborations be made on what is the "optimal" result, and, what are the technical difficulties causing the discrepancy between the results in this paper and the "optimal" result?”
>
> Our result is “near-optimal” in the sense that it is optimal up to a constant in the exponent — since one can solve the NGCA in $d^{O(k)}$ time. In fact, we believe that with the same approach but a more careful analysis, one can get a lower bound of $d^{ck}$, where c is any constant smaller than $1/8$ (and that the constant of $1/8$ cannot be improved in general).
>
> 2. Regarding the reviewer’s point:
> “For the comparison between Fact 1.6 and Theorem 1.8, why is Fact 1.6 better for $c$ close to $0$ and Theorem 1.8 better for $c$ close to $1/2$?”
>
> We believe this statement is actually not entirely accurate. Fact 1.6 uses the SQ lower bound in [DKS17], so the lower bound there in fact is roughly $\exp(O(\alpha^{-2/k}))d^{-k/(1/4-c/2)}$, where the extra $\exp(O(\alpha^{-2/k}))$ term corresponds to the chi-square distance of the one-dimensional distribution $A$ in the construction. So, if $\alpha\ll\log(d)^{-k/2}$, regardless of the choice of $c$, Fact 1.6 will always fail to give any nontrivial bound. In contrast, Theorem 1.8 can still give a nontrivial lower bound. Indeed, there are some parameter regimes where Fact 1.6 will give a quantitatively slightly better bound than Theorem 1.8 (up to a constant in the exponent). This is due to the fact that our main result is proved for any one-dimensional distribution $A$ — while the prior result of [DKS17] is proved only for one-dimensional distributions $A$ with finite chi-square distance. This allows [DKS17] to obtain a better constant factor in the exponent, but at the same time costs an extra multiplicative “chi-square distance term”.
>
> 3. Regarding the reviewer’s point:
> “Possible typo on Line 289: "$i$-th" might have been a typo of "$k$-th".”
>
> Thank you for pointing out the typo. We will address it in the final version.
>
> 4. Regarding the reviewer’s point:
> “Lemma 3.10 requires that “$d$ is at least a sufficiently large universal constant." How large does $d$ need to be in order for the results in the paper to be applicable?”
>
> We need $d$ to be at least 10 so that Lemma 3.10 is true. Note that since $d$ is the dimension in the NGCA problem, if $d$ is smaller than 10, then for any number of matching moments $k$ only depending on $d$, the problem can always be solved in $d^{O(k)}$ time (which is constant time). Therefore, the main result Theorem 3.1 is applicable for any $d$ (since if $d$ is small, then the constant is absorbed in the big-O notation).

---

> > ### Comment · Reviewer_YVEY · 2023-08-18
> > **Thanks for the response**
> >
> > Thanks for responding to my questions. I took a look at the questions raised by other reviewers and agree with some of the points raised in the reviews/discussions by g3gW and oUue. I am adjusting my confidence accordingly.

---

### Official Review · Reviewer_nJue · 2023-07-06

**Soundness:** 3 good
**Presentation:** 3 good
**Contribution:** 3 good
**Rating:** 7
**Confidence:** 4

**Summary:**




The paper discusses SQ lower bounds for Non-Gaussian component analysis. A very influential result by [DKS17] has established an SQ-lower bound suggesting d^m time as long as the non-Gaussian component's distribution A, satisfies (a) that the first m moments of A match the m moments of N(0,1) and (b) the \chi^2 between A and N(0,1) is finite.
The authors prove that the same SQ lower bound holds under only condition (a) [i.e., without assuming condition (b)]

## Correctness

From my quick check, the argument appears correct and sound.



**Strengths:**

I find the result very nice and the contribution a crucial addition to the literature.

It is quite interesting that the authors follow a direct approach, not using the "standard" SQ dimension approach introduced in [Feldman et al '2017].

**Weaknesses:**

A weakness is perhaps that the SQ lower bound technique seems to be very tailored to NGCA, as opposed to the Feldman et al 2017 result.

**Questions:**

My understanding is that by removing the \chi^2 assumption, the authors result also shows that SQ algorithms fail even when the support of A is finite. This is a quite interesting case, as it includes examples of A supported on a lattice, where lattice-based methods work in polynomial time (e.g. see [Zadik et al' Lattice-Based Methods Surpass Sum-of-Squares in Clustering], , [Diakonikolas et al' Non-Gaussian Component Analysis via Lattice Basis Reduction]). Hence, this implies a separation between SQ methods and poly-time methods in NGCA. In case that is not already known, I encourage the authors to add this interesting implication of their result.

**Limitations:**

See above.

---

> ### Author Rebuttal · Authors · 2023-08-10
>
> We thank the reviewer for their effort and positive assessment of our work. We would like to address the following questions/comments.
>
> 1. Regarding the reviewer’s point:
> “A weakness is perhaps that the SQ lower bound technique seems to be very tailored to NGCA, as opposed to the Feldman et al 2017 result.”
>
> The focus of our work is on the problem of NGCA. Specifically, we show that under the moment-matching assumption the NGCA problem is SQ-hard. We note that a wide range of learning problems have at their core “hard instances” that can be formulated as specific instances of NGCA – for an appropriate choice of the moment-matching distribution $A$; see, e.g., lines 66-74 and the associated references. As a corollary, our result implies SQ-lower bounds for all these problems.
>
> On the other hand,  [FGR+17] defines a notion of “SQ-dimension” and shows that “large SQ dimension implies SQ-hardness”. The notion of SQ dimension in that work is not sufficient for our setting. Moreover, even if the “SQ-dimension” of [FGR+17] had been sufficient (via an appropriate modification), one would need to establish that the NGCA problem has a “large SQ dimension” — which is the main technical contribution of our work. In summary, we believe that our contribution is incomparable with that of [FGR+17].
>
> 2. Regarding the reviewer’s point:
> “This is a quite interesting case, as it includes examples of A supported on a lattice, where lattice-based methods work in polynomial time (e.g. see [Zadik et al' Lattice-Based Methods Surpass Sum-of-Squares in Clustering], , [Diakonikolas et al' Non-Gaussian Component Analysis via Lattice Basis Reduction]). Hence, this implies a separation between SQ methods and poly-time methods in NGCA. In case that is not already known, I encourage the authors to add this interesting implication of their result.”
>
> The reviewer is correct. As a corollary of our result, it follows that for the case of discrete distribution $A$, efficient SQ algorithms for NGCA do not capture all polynomial-time algorithms. In particular, the LLL-based algorithm in the aforementioned works is not efficiently implementable in the SQ model. (Interestingly, the same separation holds for low-degree polynomial tests and Sums-of-Squares algorithms.) We will add an appropriate remark in the revised version of our paper.
>
> Reference:\
> [FGR+17]  V. Feldman, E. Grigorescu, L. Reyzin, S. Vempala, and Y. Xiao. Statistical algorithms and a lower bound for detecting planted cliques. J. ACM, 64(2):8:1–8:37, 2017.

---

### Author Rebuttal · Authors · 2023-08-10

We thank the reviewers for their time and effort in providing feedback. We are encouraged by the positive comments, and that the reviewers appreciated the paper for the following: (i) **importance** (YVEY), and (ii) **clarity** and **quality of writing** (YVEY, oUue). We would like to address the following question from the reviewers here.
1. “How does the main result of SQ lower bound for NGCA compare to the algorithms in [ZSWB22] and [DK22] which solve the NGCA when the hidden distribution is discrete?”

If we take the one-dimensional distribution $A$ in the NGCA to be a distribution with finite support matching the first $k$-degree moments with the standard Gaussian, where $k$ is at least a sufficiently large integer, then the SQ lower bound given by our result will be larger than the algorithmic upper bound in [ZSWB22] and [DK22]. However, as pointed out by reviewer nJue, the algorithms in [ZSWB22] and [DK22] are based on LLL lattice basis reduction which is not captured by the SQ framework, so this does not contradict our result. It is worth noting that not only the SQ model, but two other popular restricted computational models (lower-degree framework and SoS framework) also fail to capture the LLL lattice basis reduction. A comparison here can be made between the LLL algorithm and Gaussian elimination. While there is a classical exponential SQ hardness result for learning Parity functions, a polynomial-time algorithm based on Gaussian elimination can solve the problem.

We would also like to point out that the above does not imply that any NGCA family of instances with infinite chi-square distance can be solved efficiently. Importantly, LLL-based algorithms only work in the restricted setting that the support of the one-dimensional distribution $A$ is discrete/nearly discrete. For example, if the one-dimensional distribution $A$ is a mixture of a discrete distribution and a continuous distribution (as is the case in the Anti-concentration Detection problem considered in our paper), linear-algebraic algorithms will not work although the distribution has infinite chi-squared distance and the problem is believed to be hard for all efficient algorithms.

Reference: \
[DK22]  I. Diakonikolas and D. M. Kane. Non-Gaussian Component Analysis via Lattice Basis Reduction. In Conference on Learning Theory, COLT 2022, volume 178 of Proceedings of Machine Learning Research, pages 4535-4547. PMLR, 2022.\
[ZSWB22] I. Zadik, M. J. Song, A. S. Wein and J. Bruna. Lattice-Based Methods Surpass Sum-of-Squares in Clustering. In Conference on Learning Theory, COLT 2022, volume 178 of Proceedings of Machine Learning Research, pages 1247-1248. PMLR, 2022.

---

### Decision · Program_Chairs · 2023-09-21

**Decision:**

Accept (poster)

**Comment:**

This paper provides a useful technical contribution to the area of statistical-computational gaps, by proving a more general/easier-to-use SQ lower bound for the task of non-Gaussian component analysis. This problem plays a key role in lower bounds for other problems and so seems likely to be useful for future works.  Discussion between the authors and reviewers identified some useful points concerning the relation to the LLL method which the authors have promised to include in the revision. Based on this, I recommend acceptance.